# East Asia Reanalysis System (EARS)

Jinfang Yin[1], Xudong Liang[1], Yanxin Xie[1], Feng Li[1], Kaixi Hu[2], Lijuan Cao[2],

Feng Chen[3], Haibo Zou[4], Feng Zhu[5], Xin Sun[5], Jianjun Xu[6], Geli Wang[7], Ying Zhao[8],

and Juanjuan Liu[7]

[1]The State Key Laboratory of Severe Weather, Chinese Academy of Meteorological Sciences, Beijing 100081, China

[2]National Meteorological Information Center, China Meteorological Administration, Beijing 100081, China

[3]Zhejiang Institute of Meteorological Sciences, Hangzhou 310008, China

[4]Key Laboratory of Poyang Lake Wetland and Watershed Research Ministry of Education, Jiangxi Normal University, Nanchang 330022, China

[5]Inner Mongolia Meteorological Observatory, Inner Mongolia Hohhot 010051, China

[6]College of Oceanography and Meteorology, Guangdong Ocean University, Zhanjiang 524088, China

[7]Institute of Atmospheric Physics, Chinese Academy of Sciences, Beijing 100029, China

[8]School of Mathematics, Nanjing University of Aeronautics and Astronautics, Nanjing 211106, China

Submitted to *Earth System Science Data (ESSD)*

December 2022

*Correspondence to*: Xudong Liang (liangxd@cma.gov.cn)

**Abstract.** Reanalysis data plays a vital role in weather and climate study, as well as meteorological resource development and application. In this work, the East Asia Reanalysis System (EARS) was developed using the Weather Research and Forecasting (WRF) model and the Gridpoint Statistical Interpolations (GSI) data assimilation system. The regional reanalysis system is forced by the European Centre of Medium-Range Weather Forecasts (ECMWF) global reanalysis EAR-Interim data at 6-h intervals; hourly surface observations are assimilated by the Four-Dimension Data Assimilation (FDDA) scheme during the WRF model integration; upper observations are assimilated in a three-dimensional variational data assimilation (3D-VAR) mode at analysis moment. It should be highlighted that many of the assimilated observations have not been used in other reanalysis systems. The reanalysis runs from 1980 to 2018, producing a regional reanalysis dataset covering East Asia and surrounding areas at 12-km horizontal resolution, 74 sigma levels, and 3-hour intervals. Finally, an evaluation of EARS has been performed with the respect to the root mean square error (RMSE), based on the 10-year (2008-2017) observational data. Compared to the global reanalysis data of the EAR-Interim, the regional reanalysis data of the EARS are closer to the observations in terms of RMSE in both surface and upper-level fields. The present study provides evidence for substantial improvements seen in EARS compared to the ERA-Interim reanalysis fields over East Asia. The study also demonstrates the potential use of the EARS data for applications over East Asia and proposes further plans to provide the latest reanalysis in real-time operation mode. Simple data and updated information are available on Zenodo at https://doi.org/10.5281/zenodo.7404918 (Yin et al., 2022), and the full datasets are publicly accessible on the Data-as-a-Service platform of China Meteorological Administration (CMA) at http://data.cma.cn.

**Keywords:** Regional reanalysis, East Asia, Data assimilation, Multiple observations

**1. Introduction**

The East Asia Reanalysis System (EARS) project was launched by the China Meteorological Administration (CMA) in late 2014. It aimed to build a regional reanalysis system that can assimilate as much as possible multi-source observational datasets, and to establish a long-term high-resolution regional atmospheric reanalysis, which is high quality for mesoscale weather, regional climate, environment studies, and other applications. This

paper is to report the progress of the project, including the used numerical model, data assimilation, observations, and preliminary achievements. Thus, the major objectives of the present study are to (i) introduce the work we have already done; (ii) help understand and use the EARS reanalysis products; and (iii) provide guidance for repeating and extending the work in the future.

Atmospheric reanalysis data, which may serve as alternative data to actual observations, play an important role in weather and climate studies, including numerical model validation. In the past several decades, a series of atmospheric reanalysis products were produced with different goals (Wright et al., 2019); some have been widely used in theoretical studies and applied to weather and climate research to improve prediction skills and reduce hazard risks.

With the ongoing development of atmospheric sciences, high-resolution atmospheric reanalysis data are much needed. In view of this demand, a large number of high-resolution regional reanalysis products have been produced for various parts of the world (e.g., Mesinger et al., 2006; Jakob et al., 2017; Vidal et al., 2010; Usui et al., 2017; Yang et al., 2022). However, little attention has been paid to East Asia, although China's first generation of

global atmospheric reanalysis (CRA40) was released recently, with a horizontal resolution of approximately 34 km and a temporal resolution of 6 h. Only low-resolution global reanalysis products have been used for the region, including the National Centers for Environmental Prediction-Department of Energy Reanalysis version 2 (NCEP2) (Kanamitsu et al., 2002), the

European Centre of Medium-Range Weather Forecasts (ECMWF) Reanalysis-Interim

(ERA-Interim) (Dee et al., 2011), the Japanese 55-year Reanalysis (JRA55) (Kobayashi et al.,

2015), and the Modern-Era Retrospective Analysis for Research and Applications version 2

(MERRA2) (Gelaro et al., 2017). More recently, the ECMWF released the fifth generation of

its atmospheric reanalysis (ERA5) (Hersbach et al., 2020), replacing the ERA-Interim; it is a

global atmospheric reanalysis with a horizontal resolution of 0.25 degrees. Although these

global reanalysis systems have achieved great progress, their products were developed for

global coverage. They have limited regional usage due to low spatial and temporal resolution

(Chen et al., 2014). Most importantly, multiple observations over East Asia were not included

in these global reanalysis products. Consequently, the global reanalysis products are

inadequate for studying characteristics of local weather and climate in East Asia, such as

strong rainfall in the warm sector in southern China during the period from April to June (the

so-called pre-summer rainy season) (Chen et al., 2014). In view of the above-mentioned

inherent issues, it is highly imperative to develop a high spatiotemporal-resolution reanalysis

product for East Asia.

       Several regional atmospheric reanalysis datasets were produced in the past two decades,

such as the North American Regional Reanalysis (NARR) (Mesinger et al., 2006), the

High-resolution Regional Reanalysis for the European Coordinated Regional Downscaling

Experiment (CORDEX) (Bollmeyer et al., 2015; Bach et al., 2016), the Arctic System

Reanalysis (ASR) (Bromwich et al., 2017), the Bureau of Meteorology Atmospheric

high-resolution Regional Reanalysis for Australia (BARRA-R) (Jakob et al., 2017),

high-resolution regional reanalysis of Japan (NHM-LETKF) (Fukui et al., 2018), and the

regional reanalysis of Indian Monsoon Data Assimilation and Analysis (IMDAA) (Mahmood

et al., 2018). These data have been widely used for regional weather and climate studies.

Recently, Yang et al. (2022) developed a 10-year (2010-2019) regional reanalysis dataset,

focusing on the Korean Peninsula and its surrounding areas only. With the same objective, the

CMA planned a project, intending to produce high-resolution regional atmospheric reanalysis

data with high quality for mesoscale weather study and regional climate analysis over East

Asia. For this purpose, the EARS was launched in late 2014 and a 39-year (1980-2018)

reanalysis dataset is now available to the public.

This is our first open documentation of the project, on the basis of several progress

reports (e.g., Liang et al., 2020; Yin et al., 2019), which briefly describes the EARS and

documents its performance. It includes the numerical model set, data assimilation method,

assimilated observational datasets, and preliminary results. In section 2, we describe the

EARS system and the data used. In section 3, we present the preliminary results of a 10-year

(2008-2017) reanalysis dataset with validation. Finally, a summary and discussion are

provided in section 4, along with future activities and plans.

**2. East Asia Reanalysis System and Data Used**

**2.1 System Setup**

The EARS is established based on the Advance Research Weather Research and

Forecasting (WRF version 3.9.1) model (Skamarock et al., 2008) and the Gridpoint Statistical

Interpolations (GSI) data assimilation system (Hu et al., 2018). To improve the model

performance in East Asia, a series of experiments were launched for dynamic and physical

options. At present, the GIS runs in a three-dimensional variational data assimilation

(3D-VAR) mode, and much attention has been paid to the effect of assimilating each category

of observations. To date, the EARS works continuously in a 39-year run, with a cold start at

the interval of six hours.

In the EARS (Fig. 1), the WRF-ARW model grid spacing is 12 km, which covers a large

domain of an area of 10,800 km $\times$ 9,120 km (with 900 $\times$ 760 grid points); and it is centered at

(100°E, 38°N). A total of 74 sigma levels is used in the vertical, with the model top fixed at 10 hPa. The model terrain is interpolated from the 30-arc-second USGS GMTED2010, and

the land use fields are interpolated from 21-class MODIS datasets. The model physics schemes used include the following: (i) the Kain (2004) cumulus parameterization scheme; (ii) the new Thompson microphysics scheme (Thompson et al., 2008); (iii) the rapid radiative transfer model (Iacono et al., 2008) for both shortwave and longwave radiative flux calculations; (iv) the Yonsei University (YSU) Planetary Boundary Layer (PBL) scheme

(Hong et al., 2006); (v) the revised MM5 Monin-Obukhov similarity scheme for the surface layer (Janjić, 1994), and (vi) the Noah-MP land-surface scheme (Niu et al., 2011). It should be emphasized that the model set was optimized via a series of experiments covering various weather phenomena and continuous simulations (e.g., Zhang et al., 2016; Li et al., 2018; Yin et al., 2014; Yin et al., 2020).

A schematic illustrating the flow of analysis steps of the EARS is shown in Fig. 2. The WRF model integrates for 12 hours in every cold start, starting at 0000, 0600, 1200, and 1800 UTC, with hourly outputs. The ERA-Interim 0.79-degree reanalysis data at 6-h intervals are utilized as initial and boundary conditions for the cold run. Please refer to Dee et al. (2011) for detailed information on the ERA-Interim reanalysis data. At the model's initial time, the

upper-level (sounding and aircraft) observations are assimilated with the GSI system in 3D-VAR mode. During the model integration, the Four-Dimension Data Assimilation (FDDA) functions are activated by performing surface observation nudging (Reen, 2016). The required analysis data for the FDDA are obtained through the WRF's preprocessing OBSGRID module (Wang et al., 2017), using the hourly surface observations after performing data quality

control, which includes temperature, relative humidity, and horizontal winds. More specifically, observation nudging is a type of FDDA wherein artificial tendency terms are introduced during the model integration (Reen, 2016). Since it is applied at every time step,

nudging is a continuous form of data assimilation. Therefore, observations in the model

integration time window can be well ingested. Generally speaking, the differences between

the WRF model and observation are utilized to create innovations. Then, the innovations are

multiplied by various factors and added to model tendency equations. It should be noted that

observation nudging is affected by the uncertainty of the observations. Therefore, surface

observations are strictly quality controlled by the OBSGRID module (Wang et al., 2017).

The model outputs in the first six hours are considered the spin-up process, and thus not

used for research. The model outputs at the ninth and twelfth hours are used as the first guess

of the GSI, and the upper-level observations are assimilated in a 3D-VAR mode. The

upper-level observations include sounding, aircraft observation, and cloud-derived wind

vectors. The composited radar reflectivity is ingested by the way of cloud analysis to produce

the final reanalysis data. Note that model hourly outputs are also available. Unlike a

continuously operating global reanalysis system, the EARS conducts a cold start every six

hours, and the WRF model integrates 12 hours for each run. Accordingly, the model outputs

from the sixth to twelfth hours are used to produce hourly precipitation during the WRF

model integration. Before generating long-term reanalysis, the EARS was validated by

continuous simulations of the year 2014. The results indicated that the EARS performed

better in terms of atmospheric variables, and provided more mesoscale details than the

large-scale ERA-Interim reanalysis (used as background in the EARS), and its outputs can be

used for developing a long-term reanalysis product.

**2.2 Assimilated Data**

Various categories of observational data used in the EARS are listed in Table 1. The

National Meteorological Information Center (NMIC) of CMA archives all observational

datasets after performing strict data quality control. Generally speaking, several steps were

used to prepare the input observations. Firstly, the duplicate (in time and location) data reports

were merged. Secondly, all the ground-based observations were checked by climatic cut-off

values and variation ranges. Besides, internal consistency between meteorological elements

and temporal consistency were carried out. Moreover, soundings were examined based on

hydrostatic assumption, temperature lapse rate, and horizontal wind shear. The observational

data are publicly accessible at http://data.cma.cn/ (last access 8 Jan 2023). In particular, the

traditional observational datasets have been greatly improved by merging multiple data

sources, which are officially released by the NMIC. Note that many of the datasets were not

shared publicly before. Figure 3 shows spatial distributions of radiosonde and radar

observations, surface observations over land, and surface observations over the ocean. Note

that the aircraft and satellite (cloud-derived wind vector) observations are not presented due to

irregular moving trajectories depending on time and space.

Previous studies (e.g., Kawai et al., 2017; Benjamin et al., 2010; Lee et al., 2019; Rabier

et al., 2009; Ingleby et al., 2016) have confirmed that numerical model performances were

enhanced by assimilating radiosonde observations globally. Figure 4 shows the radiosonde

observations assimilated in the EARS, which has been greatly improved by combining

datasets from various databases and employing more observational data sources from China.

It can be seen that the counts of radiosonde observations show slight variation from 1980 to

2000, and then increase obviously, almost doubled by 2018. In addition to conventional

observations shared in the Global Communication System (GTS) of the World Meteorological

Organization (WMO), another 33 radiosonde stations in China are incorporated. Most

importantly, more vertical-level observations are included by merging logs of old records.

Taking the radiosonde observations of Beijing station at 0000 UTC 1 July 2016 as an example

(Fig. 5), the merged radiosonde observations show more detailed vertical structures,

compared to those in the Integrated Global Radiosonde Archive (IGRA) version 2, which is

used in the National Center for Atmospheric Research (NCAR) global reanalysis of the

Climate Forecast System Reanalysis (CFSR). Besides, radiosonde observations at both 0600 and 1800 UTC are used (Fig. 4a,c), although the observations are discontinuous and much

fewer than those at 0000 and 1200 UTC (Fig. 4b,d). Moreover, radiosonde observations from field experiments are used, including those from the Third Tibetan Plateau Atmospheric Scientific Experiment (TIPEX-III) (Zhao et al., 2018), the Southern China Monsoon Rainfall Experiment (SCMREX) (Luo et al., 2017), among others. Note that these supplementary radiosonde observations were not utilized in any global reanalysis system outside of China.

Overall, the number of radiosonde observations assimilated in the EARS has increased significantly after combining several sources, especially from 2000 to 2018.

Previous studies (e.g., Mirza et al., 2016; James et al., 2020) indicated that assimilating aircraft observations was beneficial for numerical modeling. The aircraft observations used in the EARS are provided by the NMIC after quality control (Liao et al., 2021), which is a new

product by integrating nine different data sources into the Integrated Global Meteorological Observation Archive from Aircraft (IGMOAA). Adding the datasets from the Chinese Aircraft Meteorological Data Relay (AMDAR), the observation count has increased significantly, compared to that of the IGMOAA. The integrated data were officially released by the NMIC and are updated in real-time at the Data-as-a-Service platform

(http://data.cma.cn/). Generally speaking, aircraft observations were rare in the early days, and these observations have increased greatly from $2\times10^3$ in 2005 to $7\times10^3$ in 2018 (Fig. 6). Although the aircraft observations are hourly, there are large differences at different moments of the day. One can see that the count of aircraft observations in the daytime is much larger than that in the nighttime, though the count of aircraft observations in the nighttime has

slightly increased since 2005.

For surface observations over land (Fig. 3b and Fig. 7a), besides those from the National Centers for Environmental Prediction (NCEP) Global Data Assimilation System (GDAS) and

the National Climatic Data Center (NCDC) Integrated Surface Database (ISD), more than 2440 surface observations from the CMA are added. Detailed processing of the datasets can

be found in Jiang et al. (2021). Note that only a small portion (less than 300 stations in the early days and nearly 400 stations at present) of the surface observations is shared in the GTS. The observations over the sea surface are combined with the International Comprehensive Ocean-Atmosphere Data Set (ICOADS) (Fig. 3c). After the combination, the ocean-based observation count used in the EARS increases by approximately 32% in total, compared to

the ICOADS. As shown in Fig. 7b, surface observations have increased significantly, especially since 2000. All the hourly surface observations (over land and ocean) are further quality controlled by using the OBSGRID module provided by the WRF Variational Data Assimilation (WRFDA); and the outputs in ASCII format are used for observation nudging during the WRF model integration. We pointed the readers to Skamarock et al. (2008) for

more details. Similarly, all the upper-level traditional observational datasets are quality controlled by using the OBSPROC module, and then written in the prepBUFR format for GSI assimilation in a 3D-VAR mode.

One of the main features of the EARS is its emphasis on radar data assimilation. All weather radar observations over China are used in the EARS (Fig. 3d). Radar observations

have increased rapidly from 80 stations in 2008 to over 190 stations in 2018 (Fig. 8). Note the radar observations show obvious seasonal various because some radars were switched off in cold seasons due to the absence of weather processes. To obtain high-quality-controlled radar observational data, much attention has been paid to the preprocessing of raw radar data. A major issue of the radar observations is the non-meteorological echo, which has direct

influences on the cloud hydrometeors in the GSI cloud analysis processes. In view of this issue, much effort has been devoted to removing isolated non-meteorological echoes and ground clutters from the radar data (Zou et al., 2018), which makes quality-controlled radar

data more accurate (Wu et al., 2018). After quality control, all radar observations at the same time are utilized to generate mosaic products in BUFR format, which can be inserted into the

GSI cloud reanalysis module. Detailed information about radar data processing and remapping can be found in Zou et al. (2014). The cloud analysis module in the GSI came from the Advanced Regional Prediction System (ARPS) (Hu and Xue, 2007), and can be further traced to the Local Analysis and Prediction System (LAPS) (Albers et al., 1996). In fact, quality-controlled radar observations are also an important part of the reanalysis data,

which can be used for weather and climate studies, as well as numerical model validation. Despite considerable effort expended in processing radial wind, the radial wind has been not assimilated at present as more work is required.

Another feature is the application of the cloud-derived wind vector datasets from Fengyun-2 geostationary meteorological satellites. The cloud-derived wind vector appears

with a count of nearly $6.0 \times 10^4$ (Fig. 9). Note that the data have been strictly quality controlled and widely applied in daily operational numerical weather prediction in China; thus these datasets can be applied directly in the EARS.

**Table 1**. Observational data used in EARS. Those data are publicly accessible at

http://data.cma.cn/ (last access 8 Jan 2023)

| Data | Variable | Starting year |
| --- | --- | --- |
| Radiosonde | Pressure, temperature, wind, and moisture | 1980 |
| Aircraft | Temperature and wind | 1980 |
| Surface[1] | Pressure, temperature, wind, and moisture | 1980 |
| Radar | Radar reflectivity | 2008 |
| Satellite | Cloud-derived wind vector | 2008 |

---

[1] Including the surface observations over ocean.

## 2.3 Validation Data and Method

The performance of the EARS is assessed by comparing it with observations and with the large-scale forcing of the ERA-Interim reanalysis (used as the background of the EARS). A 10-year good quality and stable quantity of observations are selected for the assessment. For comparison with station observations, results from the EARS and ERA-Interim are interpolated onto the stations using the nearest neighbor interpolation via the Model Evaluation Tools (MET), which is developed by the Developmental Testbed Center (DTC) of the U.S. (Newman et al., 2022). Following the operational model evaluations of the CMA, we use a total of 2,423 surface observational stations as the reference for the reanalysis data validation. The observations over the Qinghai-Tibet Plateau and its surrounding areas are much sparser, compared to those over the other regions (Fig. 10a). Similarly, a total of 120 radiosonde data over China are used to evaluate high-level variables (Fig. 10b). As has been stated above, the radiosonde observations are mainly obtained at 0000 UTC and 1200 UTC, and the measurements include temperature, (relative) humidity (or dew-temperature), air pressure, horizontal wind speed, and direction. It should be noted that the present validation is based on the observations from CMA. Although the EARS covers a large area, only limited observations out of China were obtained by the GTS. Comparatively speaking, the density of observations is much higher in China than that outside of China. Besides, the performance of observations in China is at a comparable level because of the same (at least equivalent) observational instruments and methods. Moreover, the observations in China were quality controlled using the same methods. Therefore, the observations in China were used in the validation. We welcome more validation from others with observations outside of China as much as possible.

The quality of the regional reanalysis is also compared to one-month (July 2016) continuous radiosonde observational data, which were obtained from a field experiment in the

central Taklimakan Desert, Xinjiang Uygur Autonomous Regions, China. The central

Taklimakan Desert is far from other observation sites, where the assimilated observations

have little influence on the reanalysis data. Note that these data have not been applied to any

weather forecasting or reanalysis systems (including global and regional systems), which

should be an excellent source of independent observations (Huang et al., 2021). The

radiosonde observational station (marked with a black star in Fig. 10b) is located at (83.63°E,

39.04°N), 1,099.3 m above the sea level. The radiosonde observational data were collected

four times at 0000, 0600, 1200, and 1800 local standard time (LST, = UTC + 6) a day in July

2016, using the Global Positioning System (GPS)-based radiosonde. One of the advantages of

the radiosonde observation is its high vertical resolution, which is achieved by high frequency

(at intervals of one second) data acquisition during balloon ascending.

To assess the new EARS data, we compare them with surface and radiosonde

observations in terms of root mean square error (RMSE), given by

$$RMSE = \sqrt{\frac{1}{N}\sum_{i=1}^{N}(F_i - O_i)^2} \, , \qquad (1)$$

where $N$ is the total number of all observations; $F_i$ and $O_i$ denote reanalysis data and

observation, respectively.

We use RMSE(EARS) and RMSE(ERA-I) to represent the RMSEs for the regional

EARS and global ERA-Interim, respectively. The fractional percentage improvement ($I$) of

the RMSE can be defined as follows:

$$I(\%) = \frac{RMSE(\text{ERA-I}) - RMSE(\text{EARS})}{RMSE(\text{ERA-I})} \times 100\% \, . \qquad (2)$$

Accuracy is perhaps the most widely used objective validation method for quantitative

precipitation forecasts. Following the MET verification measures for categorical

(dichotomous) variables, we employ a table of 2 × 2 contingency (Table 2). The accuracy of

precipitation forecast is defined by

$$Accuracy = \frac{hits + correct\ negatives}{total}.$$
(3)

The accuracy ranges from 0 to 1, and a perfect forecast would have an accuracy of 1.

**Table 2**. Contingency table for categorical (dichotomous) variables.

| | | observation | | |
|---|---|---|---|---|
| | | yes | no | total |
| reanalysis | yes | hits | false alarms | forecast yes |
| | no | misses | correct negatives | forecast no |
| | total | observed yes | observed no | total |

**3. Results**

**3.1 Performance of Surface Field**

Figure 11 compares 10-year-averaged RMSEs of surface variables from the EARS and ERA-Interim using box-percentile plots. For the EARS, the averaged RMSEs of surface pressure ($P$), temperature ($T$), specific humidity ($Q$), zonal wind ($U$), meridional wind ($V$), and wind speed ($WS$) are 14.11($\pm$19.22) hPa, 2.05($\pm$1.43)°C, 1.18($\pm$0.28) g kg$^{-1}$, 1.76($\pm$0.69) m s$^{-1}$, 1.95($\pm$0.63) m s$^{-1}$, and 2.06($\pm$0.58) m s$^{-1}$, respectively. The ERA-Interim has larger averaged RMSEs of $P$, $T$, $Q$, $U$, $V$, and $WS$, which are 24.34($\pm$27.17) hPa, 2.25($\pm$1.43)°C, 1.33($\pm$0.35) g kg$^{-1}$, 1.98($\pm$0.58) m s$^{-1}$, 2.35($\pm$0.70) m s$^{-1}$, and 2.42($\pm$0.51) m s$^{-1}$, respectively. In terms of the RMSE, the EARS performs much better, with respective improvement percentages of 42.01%, 8.82%, 11.28%, 11.37%, 16.96%, and 14.75% for $P$, $T$, $Q$, $U$, $V$, and $WS$, respectively. Generally speaking, $P$ has the largest improvement, followed by $V$, $U$, and $Q$; and $T$ has the smallest improvement. Similarly, the 25th, 50th, and 75th percentiles show obvious improvements.

In terms of the RMSE, we have noted that the EARS shows an obvious improvement in

335 the surface meteorological fields over East Asia, compared to the ERA-Interim (Fig. 11). According to the statistical results, $P$ has the largest improvement percentage of the RMSE, followed by $V$; and $T$ has the smallest improvement. The smaller RMSE is mainly attributable to the data assimilation of a large number of observations. Note that the optimized WRF model, focusing on East Asia with a high horizontal resolution of 12 km and 74 sigma levels

340 in the vertical, is also beneficial to the smaller RMSE. As has been stated above, the WRF model was tested and verified in various aspects by paying attention to dynamic and physical options, and to the observation nudging parameters (Yin et al., 2018). According to our previous tests with the optimized WRF model, the downscaling results performed better than the ERA-Interim, which provides good background conditions for the GSI data assimilation

345 and the subsequent reanalysis data. Previous studies (e.g., Andrys et al., 2015; Gao et al., 2022; Qiu et al., 2017) also indicated that significant performances have been gained in WRF downscaling at a high resolution.

  Figure 12 shows spatial distributions of the averaged RMSEs of $P$ for the EARS and ERA-Interim, and for their differences. Clearly, the spatial distribution for the EARS is

350 similar to that of the ERA-Interim. Given the spatial distribution, there is a smaller RMSE over eastern China, which is beneficial from the ingestion of dense surface observations. Note that the higher resolution of the complex terrain over western China has positive contributions to the model results, except for limited observations over this region. More specifically, both EARS and ERA-Interim show large RMSEs over western China, especially along the east

355 side of the Qinghai-Tibet Plateau. The spatial distributions for other surface variables are also generated, although there are not presented here. Please refer to the supplementary for more results. In general, the EARS has similar improvements to $P$ in $T$. Concerning $Q$, both EARS and ERA-Interim have obvious RMSEs over southern China, which may be related to a large amount of available water vapor in this region. The EARS shows a similar spatial distribution

of RMSE of $U$ to that of the ERA-Interim, while obvious differences can be found in the

RMSE of $V$. A large RMSE belt of $U$ is in the northeast toward the Tibet Plateau. The EARS

reduces RMSE in $V$ significantly, indicating that the quality of $V$ is improved considerably.

The EARS alleviates the RMSE over values 2.5 m s$^{-1}$ at most stations. Note that the EARS

has larger RMSEs in the wind field (both $U$ and $V$) over western China than the ERA-Interim.

This may be related to complicated dynamics associated with the Tibet Plateau, land

processes, or/and poor quality of observations over this region; and attention is required to

understand the shortcoming. For $WS$, the improvement shows a similar pattern to that in $V$. In

terms of RMSE, the EARS performs better than the ERA-Interim at most stations, although

the EARS has poor quality at some stations.

**3.2 Upper-Level Fields**

Figure 13 shows the mean RMSEs of vertical profiles of $U$, $V$, $T$, and $Q$, verified against

120 radiosonde observations over China during 2008-2017 (Fig. 10b). Generally speaking, the

EARS shows much smaller RMSEs than the ERA-Interim at all levels, although both show

similar vertical distributions. The RMSEs for the ERA-Interim are nearly twice as large as

those of the EARS, except for the RMSEs of $Q$ at the upper levels. More specifically, the

RMSEs of $U$ and $V$ for the EARS are nearly 1 m s$^{-1}$ throughout the vertical column, while

those of the ERA-Interim are mostly larger than 2 m s$^{-1}$. Also, the RMSEs of $U$ and $V$ for the

EARS show slight variation in vertical, while the RMSEs of ERA-Interim are large at the

lower and upper levels and small at the middle levels. As for the RMSEs of $T$, the RMSEs of

the EARS are within 0.9°C; and the RMSEs of the ERA-Interim are less than 1.5°C. Both

reanalysis products show large RMSEs at the lower level near the ground; and the RMSEs for

$T$ decrease first with increasing height, bottomed out near 400 hPa. The second-largest

RMSEs for $T$ occur at the higher level of 100 hPa. The large RMSEs for $T$ at the upper levels

mainly result from limited radiosonde observations. Besides, the interactions between the

troposphere and stratosphere may have some impact on the accuracy of the reanalysis products. The RMSEs for $Q$ decrease rapidly with increasing height and approach zero near 200 hPa. It should be pointed out that the small RMSEs at the upper levels mainly result from a very low value of $Q$, rather than from having a good performance at these levels. In view of the vertical profiles of the EARS verifying against radiosonde observations given in Fig. 13,

the EARS is considerably better than the ERA-Interim. The RMSEs for EARS are almost half as small as that of the ERA-Interim. The RMSEs of $U$, $V$, and $T$ for the EARS are considerably smaller than those of the ERA-Interim. At upper levels above 500 hPa, the RMSEs of $Q$ for both EARS and ERA-Interim are similar in magnitude, while the former shows a smaller value than the latter. As stated in Mesinger et al. (2006), the reanalysis data

are influenced by both the estimate of the background and observation error covariances.

### 3.3 Rainfall

Despite several objective verification methods for modeling quantitative rainfall amounts, systematic assessment of simulated rainfall performance remains difficult. Consequently, a simple comparison between the EARS and ERA-Interim is given here in terms of the

accuracy of 3-h accumulated rainfall. Please refers to Yang et al. (2023) for detailed analyses of the simulated rainfall properties of the EARS. Figure 14 shows the accuracy of 3-h rainfall for both EARS and ERA-Interim. Although the accuracy shows slight diurnal variation, both EARS and ERA-Interim have high averaged accuracies of over 0.5 and show good performances from early morning (2100 UTC) to the midday on next day (0300 UTC). The

EARS has higher rainfall accuracy than the ERA-Interim at all times. For an overview (i.e., mean), the EARS provides a higher total mean accuracy of 0.61, with 0.56 for the ERA-Interim. Note that the improvements vary from 4.53% to 16.18%, with the averaged improvement percentage of the accuracy being 8.99%. We also calculated the equitable threat score for 3-h accumulated rainfall (not shown). For rainfalls above 20.0 mm, the EARS is

much better than the ERA-Interim, indicating that the EARS has better capability to
reproduce heavy rainfall over East Asia, especially for 3-h accumulated rainfall that is over 50
mm. Note that the ERA-Interim cannot capture 3-h accumulated rainfall that is over 70 mm,
which may be caused by the global model's low resolution of nearly 79 km.

Figure 15 shows the spatial distribution of averaged 3-h accumulated rainfall accuracy
for the EARS and ERA-Interim in 2008-2017. Clearly, both EARS and ERA-Interim have
high forecast capability for precipitation over central China with rainfall accuracy of over 0.6,
followed by southern and eastern China (Figs. 15a, b). The low accuracy (less than 0.4)
mainly appears in western China, especially over the Qinghai-Tibet Plateau. The results are
consistent with previous studies and with operational predictions (e.g., Mao et al., 2010;
Zhang et al., 2021; Zhao et al., 2018). In general, the EARS has better performance on the 3-h
scale at most stations than the ERA-Interim (Fig. 15c), although the EARS has less accuracy
at some stations. The EARS with more local observations is probably the main reason for its
better performance, and the benefit of the optimized WRF model with a high resolution of 12
km may be another reason. The results indicate that the EARS would be more suitable for
investigating precipitation over East Asia.

**3.4 Features in Lower Levels over Central Taklimakan Desert**

Figure 16 shows the diurnal variation of observed and simulated vertical thermal
structures in the lower levels (0.6 km above the ground). From the observations, obvious
transitions exist in the thermal structure. More specifically, there is an inversion layer near the
surface in the nighttime, while a sub-adiabatic or superadiabatic layer occurs in the daytime.
The transition, from stable to convective and back to stable condition, is consistent with the
diurnal variation of solar radiation (Yin et al., 2021). In general, both the EARS and
ERA-Interim are able to reproduce similar diurnal transitions as in the observations. Although
there are some differences between the reanalysis products and observations, the transitions,

from stable to convective and back to stable condition, are consistent with the observations. Specifically, the EARS is closer to the observations, compared to the EAR-Interim. For instance, the EARS captures the obvious inversion at 0600 local standard time (LST), while the ERA-Interim underestimates the inversion.

    Figure 17 compares the averaged profiles of observed horizontal wind and those of the

reanalysis products. An obvious directional shift from northeasterly to westerly appears nearly 2.6 km above the ground on average. In fact, the altitude of the wind directional shift exhibits noticeable diurnal fluctuation, bottomed out at 0600 LST and peaked at 1800 LST with altitudes near 2.0 and 3.4 km, respectively. Note that the horizontal wind speed decreases and then increases with increasing height due to vertical wind shear. Compared to the observations,

both EARS and ERA-Interim capture the principal vertical wind profile patterns over the central Taklimakan Desert. However, the diurnal variation of wind profiles is slightly underestimated by the EARS, while the ERA-Interim completely misses the diurnal fluctuation. Besides, the ERA-Interim underestimates wind speed near the surface; it seems that the ERA-Interim reanalysis system cannot well describe the near-surface thermodynamic

processes.

## 4. Conclusions and future outlook

    We present a detailed report about the EARS, including 39-year (1980-2018) high-resolution regional reanalysis datasets over East Asia that show major improvements over the global reanalysis in both spatial resolution and accuracy. The qualities of the

reanalysis dataset were verified based on surface observations and radiosonde observations from 2008 to 2017, as well as radiosonde observations from field experiments in July 2016. Regarding resolution, the 12-km grid is much higher than those of global models. For accuracy, both near-surface and upper-level fields are closer to the observations than the global reanalysis ERA-Interim.

The EARS is established based on the WRF-ARW model and GSI data assimilation system. To improve the EARS's performance in East Asia, a series of experiments have been conducted for selecting dynamic and physical options. For the GSI, much attention was paid to the improvements of assimilating each category of observations. The EARS started cold runs every day, starting at 0000, 0600, 1200, and 1800 UTC with the ERA-Interim

0.79-degree analysis data at 6-h intervals as initial and boundary conditions. The WRF model was integrated for 12-h each time with hourly outputs, and hourly surface observations were ingested by performing surface observation nudging. The model outputs at the ninth and twelfth hours are used as the first guess of the GSI; the upper-level observations were assimilated in a 3D-VAR mode; and mosaic radar reflectivities were ingested by cloud

analysis.

      An important feature of the EARS is the use of a large number of observations from CMA. Compared with IGRA version 2, more than 33 operational radiosonde observations over China were used. Besides, more radiosonde vertical-level observations were included by merging logs of old records. Moreover, radiosonde observations from field experiments over

China were also employed. A large number of aircraft observations and surface (over land and sea) hourly observations over China were utilized. Note that only a small portion of the observations has been shared in the GTS. Another characteristic is the application of over 200 Doppler radar observations and the cloud-derived wind vector datasets from Fengyun-2 geostationary meteorological satellites.

To the present, 39-year (1980-2018) reanalysis data have been achieved. To assess the EARS data, 10-year (2008-2017) data were compared with surface and radiosonde observations in terms of RMSE. The results show substantial improvements in the EARS, compared to the ERA-Interim reanalysis over East Asia. The better performance of the EARS is mainly attributable to the data assimilation of a large number of observations. In addition,

the optimized WRF model, focusing on East Asia with a high resolution of 12 km and 74

sigma levels, is also beneficial to the high quality of the EARS. It should be noted that the

present validation is based on the observations from CMA. Although the EARS covers a large

area, only limited observations were obtained by the Global Communication System (GTS).

We welcome more validation from others with observations outside of China as much as

possible.

To date, We are fully occupied with EARS development and data generation. The EARS

data was verified against both surface and sounding observations. The results were also

compared with its parent—the ERA-Interim. At present, comparisons with other global

reanalysis have not been undertaken. As far as we know, the assessment of reanalysis data is a

complex and systematic task. Therefore, we expect more scholars to evaluate EARS data from

different aspects, such as the performance in reproducing weather systems (e.g., Gong et al.,

2022), daily variation in precipitation (e.g., Li et al., 2017), and others, as well as comparisons

among different reanalysis (e.g., Yang et al., 2023). In the future, we will further inspect the

regional high-resolution data against the observations from the Second Tibetan Plateau

Scientific Expedition and Research (STEP) program, in particular using it in high-resolution

studies over the East Asian monsoon region. Besides, radar retrieval horizontal wind, which is

retrieved by an improved version of the Integrating Velocity-Azimuth Process (IVAP)

method (Liang et al., 2019), will be ingested by performing upper-level observation nudging.

Most importantly, from 2019 onward, we will shift to using the ERA5 products as initial and

boundary conditions for the WRF model. Besides, the intensive surface observations

(exceeding 80,000) over China after strict quality control will be introduced in the surface

observation nudging. The EARS will run in real-time operation mode to provide the latest

reanalysis data with approximately a 5-day lag (depending on the availability of the ERA5

data).

 **Appendix A: Abbreviations**

**Table A1** List of abbreviations used in the paper.

| Abbr. | Term |
|---|---|
| 3D-VAR | Three-dimensional variational data assimilation |
| AMDAR | Aircraft Meteorological Data Relay |
| ASR | Arctic System Reanalysis |
| BARRA-R | Bureau of Meteorology Atmospheric high-resolution Regional Reanalysis for Australia |
| CMA | China Meteorological Administration |
| CFSR | Climate Forecast System Reanalysis |
| CORDEX | Coordinated Regional Downscaling Experiment |
| CRA40 | China's first generation of global atmospheric reanalysis |
| EARS | East Asia Reanalysis System |
| ECMWF | European Centre of Medium-Range Weather Forecasts |
| ERA5 | ECMWF fifth generation of its atmospheric reanalysis |
| ERA-Interim | ECMWF ERA-Interim reanalysis |
| FDDA | Four-Dimension Data Assimilation |
| GDAS | Global Data Assimilation System |
| GSI | Gridpoint Statistical Interpolations |
| GTS | Global Communication System |
| ICOADS | International Comprehensive Ocean-Atmosphere Data Set |
| IGMOAA | Integrated Global Meteorological Observation Archive from Aircraft |
| IGRA | Integrated Global Radiosonde Archive |
| IMDAA | Indian Monsoon Data Assimilation and Analysis |
| ISD | Integrated Surface Database |
| JRA55 | Japanese 55-year Reanalysis |
| MERRA2 | Modern-Era Retrospective Analysis for Research and Applications version 2 |
| MET | Model Evaluation Tools |
| NARR | North American Regional Reanalysis |
| NCAR | National Center for Atmospheric Research |
| NCDC | National Climatic Data Center |
| NCEP | National Centers for Environmental Prediction |
| NCEP2 | National Centers for Environmental Prediction-Department of Energy Reanalysis version 2 |
| NHM-LETKF | high-resolution regional reanalysis of Japan |
| NMIC | National Meteorological Information Center (NMIC) of CMA |
| RMSE | Root mean square error |
| WMO | World Meteorological Organization |
| WRF | Weather Research and Forecasting |
| WRFDA | WRF Variational Data Assimilation |

**Data Availability**

A Digital Object Unique Identifier (DOI: https://doi.org/10.5281/zenodo.7404918) is

available for the EARS reanalysis data, which provides comprehensive and up-to-date information about EARS and sample data. The 39-year EARS data reported in this work are available at 3-h intervals, starting at 00 UTC from 1980 to 2018. The database format is GRIB version 1 and the total volume of the data files is 54.6 TB. The GRIB files are hosted at the CMA Data-as-a-Service platform (http://data.cma.cn/) as their total volume exceeds the

volume that could be provided by Zenodo (Yin et al., 2022). Simple data and updated information are available on Zenodo at https://doi.org/10.5281/zenodo.7404918 (Yin et al., 2022), and the full datasets are publicly accessible on the Data-as-a-Service platform of China Meteorological Administration (CMA) at http://data.cma.cn. In general, users can obtain comprehensive and up-to-date information about EARS and sample data in Zenodo., and all

data can be downloaded from the CMA Data-as-a-Service platform (http://data.cma.cn/). The data can be obtained in the form of a hard disk copy by contacting the authors at present and will be accessed freely at this location soon. The EARS 3-h data on pressure levels and hourly precipitation data are available in GRIB format, which can be used as a model (e.g., the WRF) forcings. Owning to the large amounts of data, more variables, and datasets on the 74 model

(sigma) levels can also be obtained by contacting the authors.

**Author contributions.** X. Liang proposed the main scientific ideas and contributed to radar data processing. J. Yin contributed supplementary ideas and produced the 39-year regional reanalysis and validation. F. Chen and J. Yin developed the reanalysis system, J. Yin, X. Liang,

and Y. Xie analyzed the simulations and wrote the manuscript. K. Hu, L. Cao, and F. Li contributed to the preparation of observational datasets and reanalysis management. H. Zou completed the composite radar reflectivity for complex cloud analysis. G. Wang and Y. Zhao launched the simulations for WRF model optimization. J. Liu and J. Xu conducted data assimilation experiments. F. Zhu and X. Sun prepared the programs for validation.


**Competing interests.** The authors declare that they have no competing interests.

**Acknowledgments.** This work was conducted by using the PI-Dawning supercomputer system provided by the China Meteorological Administration (CMA). The authors are

grateful to the European Centre of Medium-Range Weather Forecasts (ECMWF) for providing the global reanalysis EAR-Interim data (https://www.ecmwf.int/en/forecasts/datasets/reanalysis-datasets/era-interim). We thank Dr. Zhiquan Liu at the National Center for Atmospheric Research (NCAR) and Prof. Zijiang Zhou at the National Meteorological Information Center (NMIC) of CMA for their helpful

discussions. All the figures are generated by NCAR Command Language (NCL), which is available at http://dx.doi.org/10.5065/D6WD3XH5 (last accessed on 20 September 2022).

**Financial support.** This study is jointly supported by the National Key Research and Development Program of China (2017YFC1501800), Second Tibetan Plateau Scientific

Expedition and Research (STEP) Program (2019QZKK010402), National Natural Science Foundation of China (42075083), and S&T Development Fund of Chinese Academy of Meteorological Sciences (2023KJ047).

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

# Figures

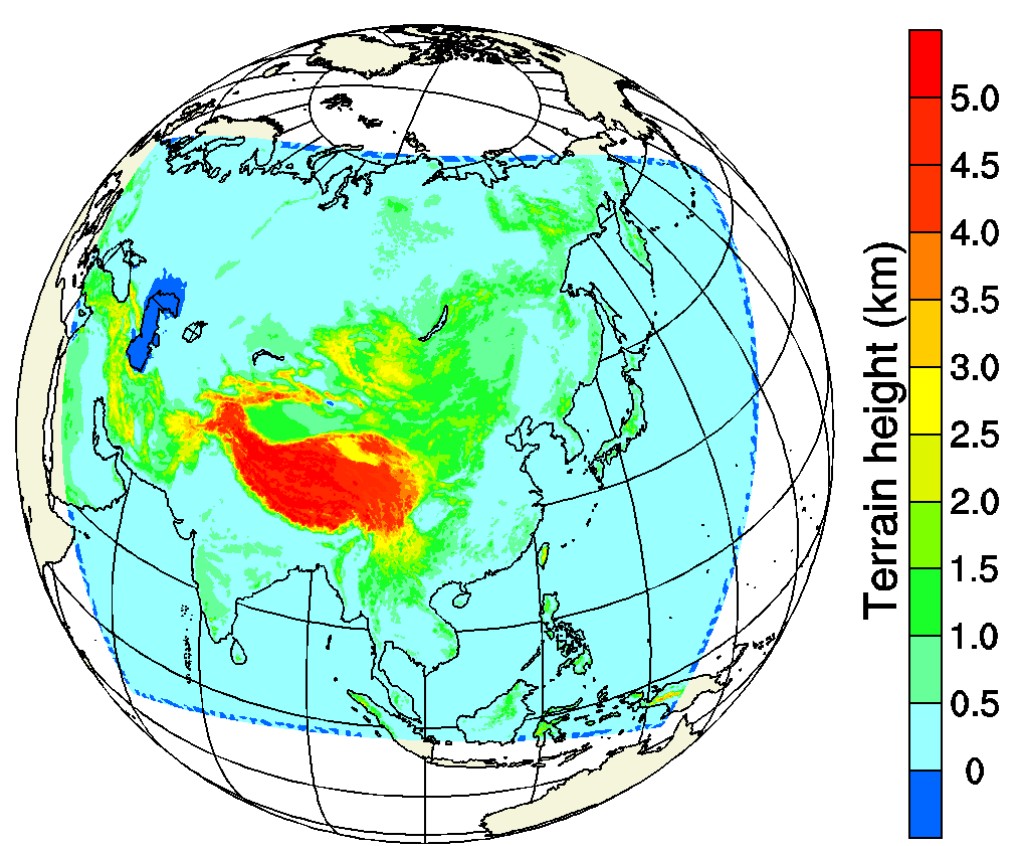

**Fig. 1** The East Asia Reanalysis System (EARS) domain in the WRF model, and its 12-km topography. The shading indicates the height of the terrain in the model.

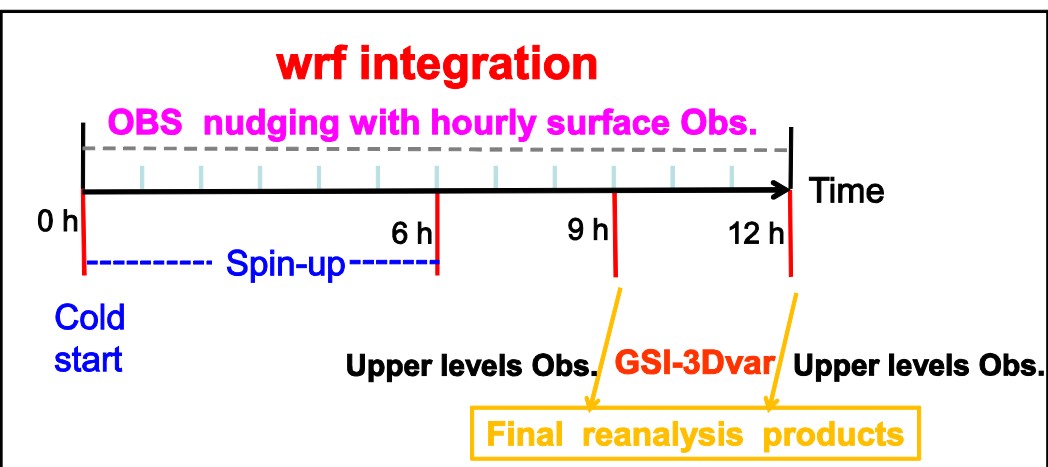

**Fig. 2** Schematic illustrating the flow chart of analysis steps of the EARS. See text for details.

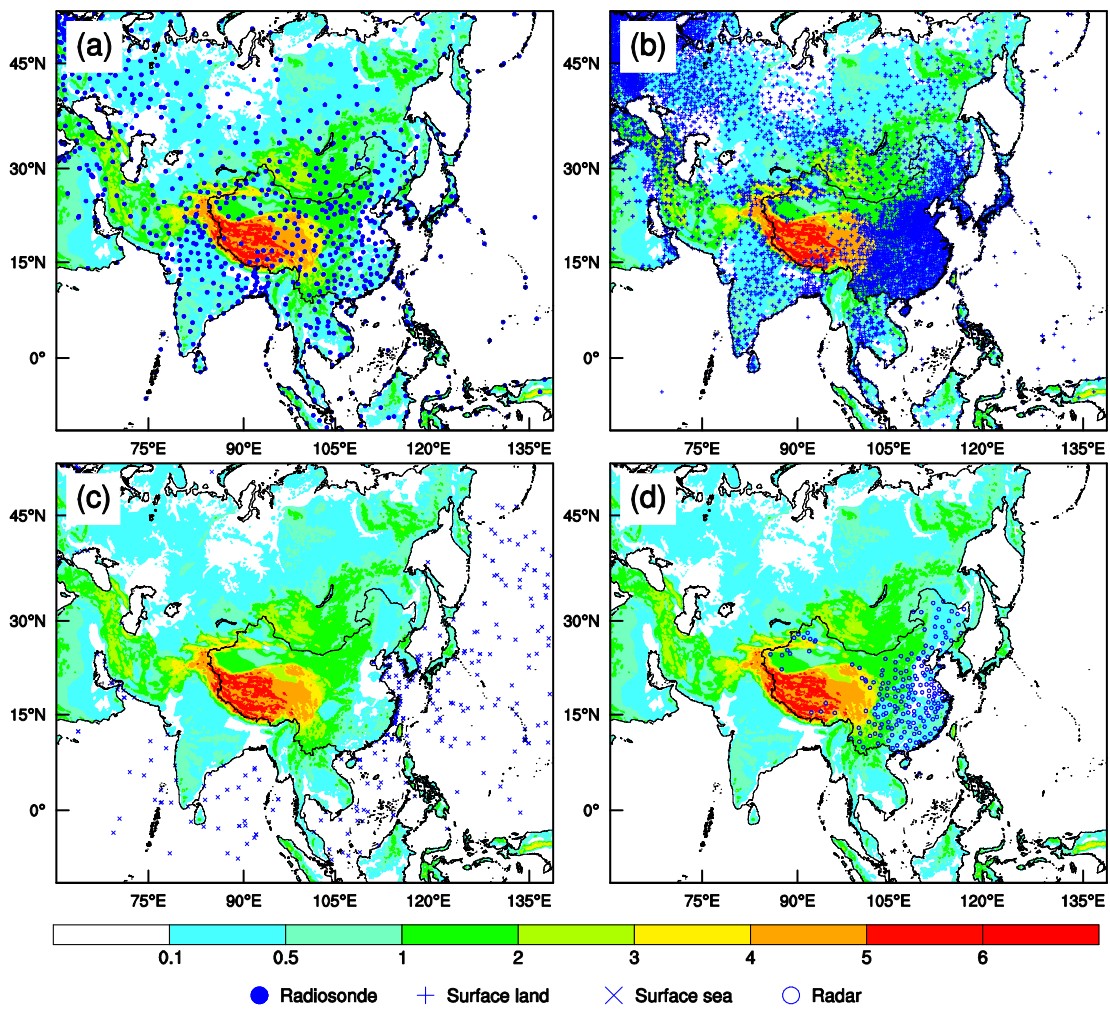

**Fig. 3** Spatial distributions of (a) radiosonde, (b) land, (c) sea, and (d) radar observations for EARS.

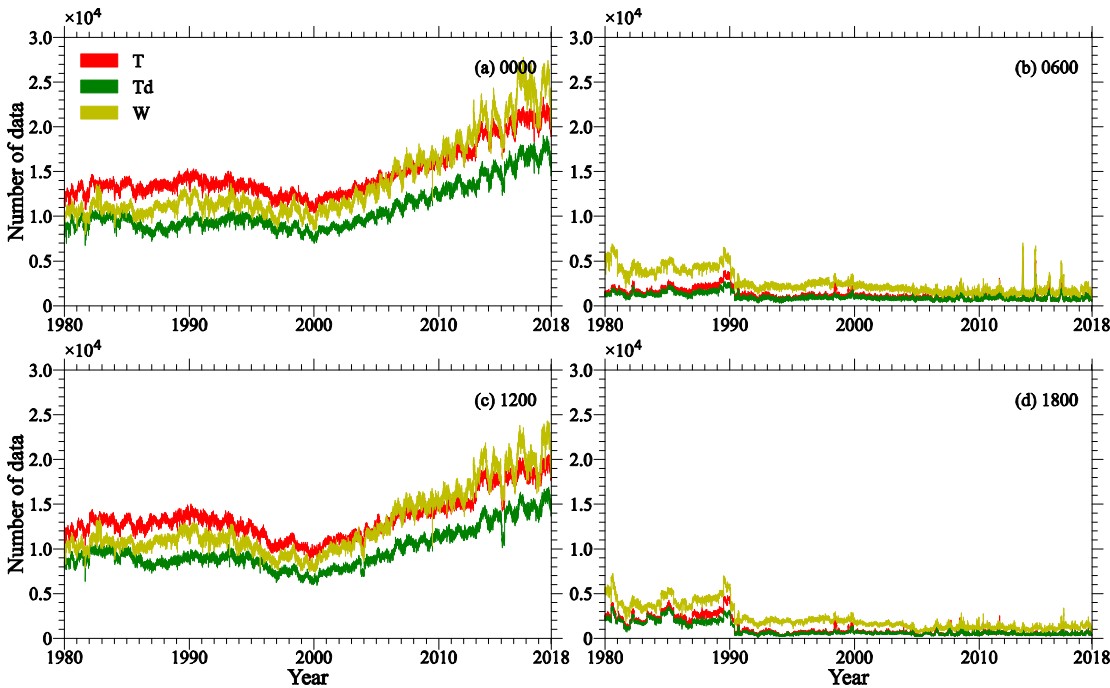

**Fig. 4** Counts of radiosonde observations assimilated in EARS at (a) 0000, (b) 0600, (c) 1200, and (d) 1800 UTC. $T$, $T_d$, and $W$ denote temperature, dew temperature, and horizontal wind, respectively.

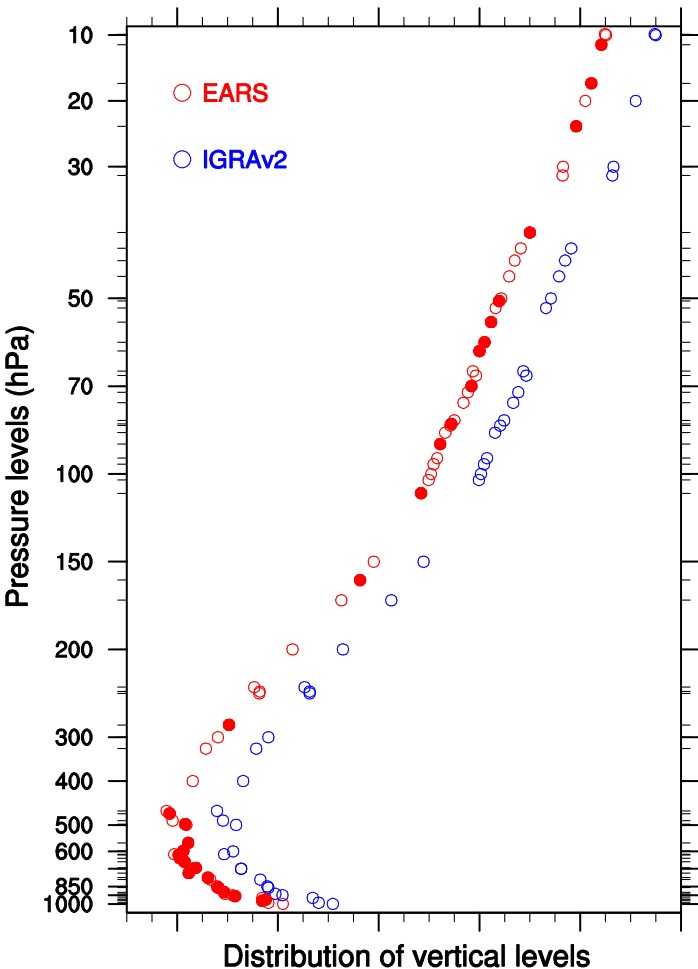

**Fig. 5** Comparison of merged radiosonde (red) used in EARS and the Integrated Global Radiosonde Archive version 2 (IGRAv2, blue) at Beijing station (54511) at 0000 UTC 1 July 2016. The red dots represent the newly merged from paper-based records that have never been used in any atmospheric reanalysis system outside of China. Note that the IGRAv2 profile was shifted to the left to avoid overlaying the two datasets. The two profiles are perfectly overlapped except for newly added observation points.

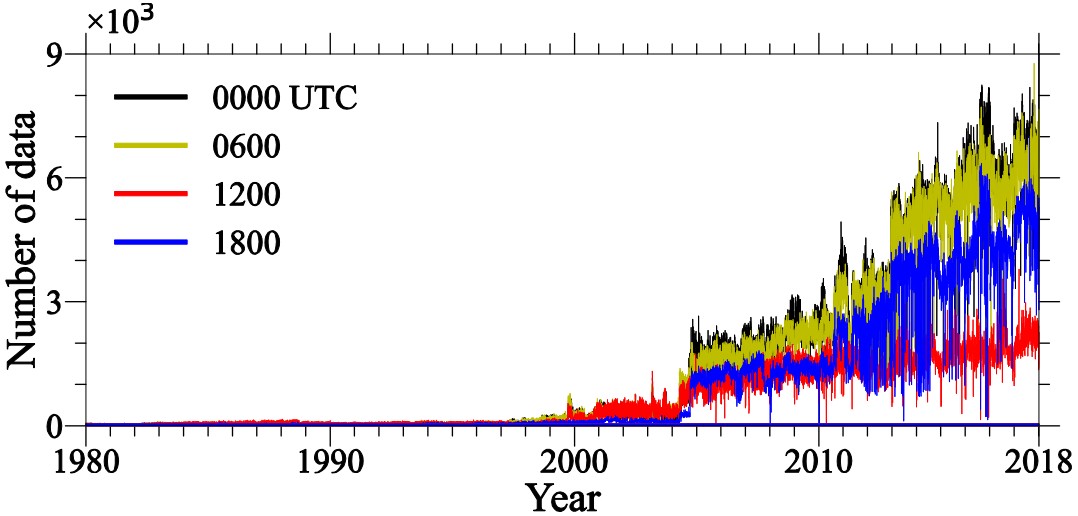

**Fig. 6** Counts of aircraft observations used in EARS at 0000, 0600, 1200, and 1800 UTC.

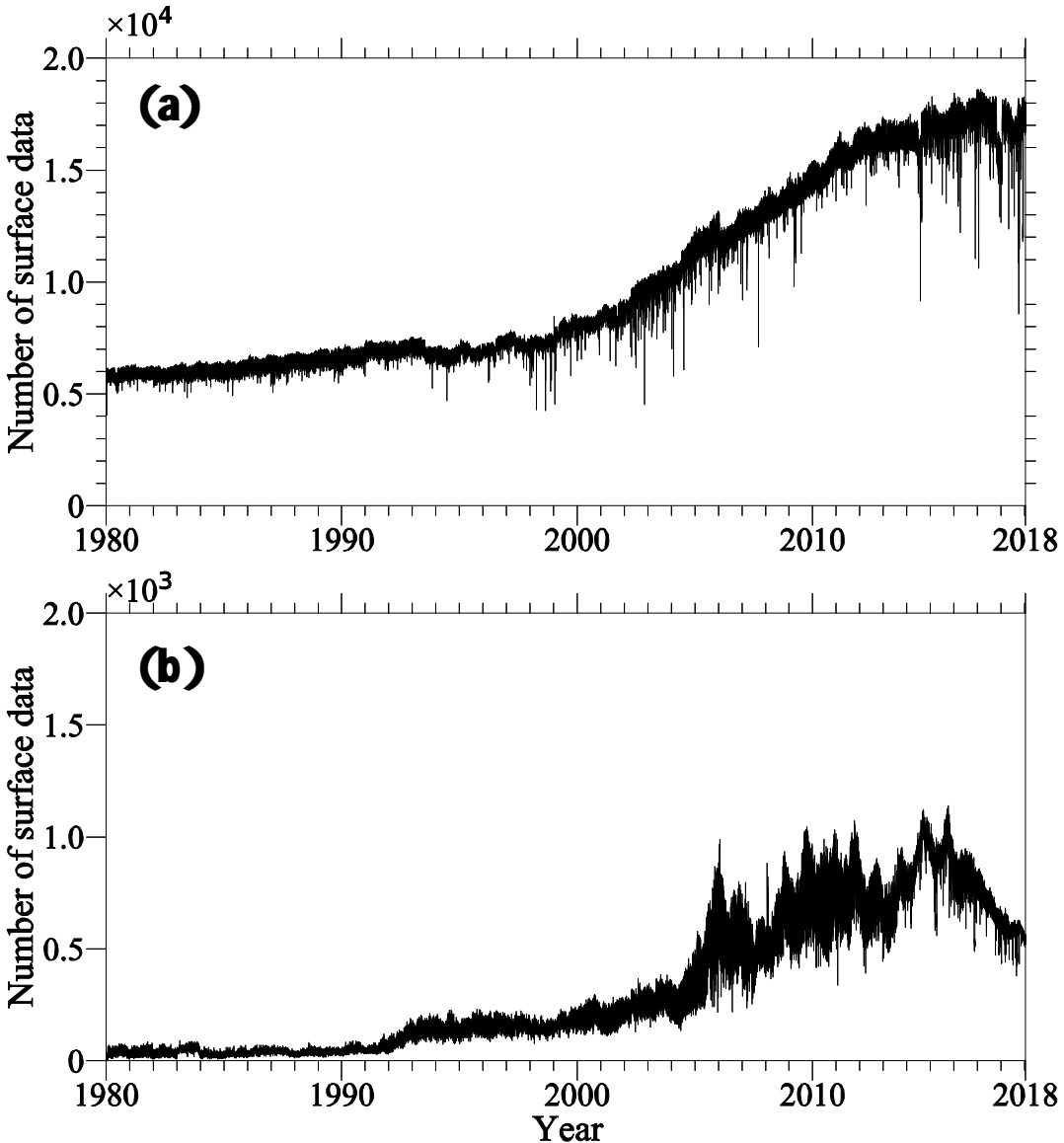

**Fig. 7** Counts of the surface observations over (a) land and (b) sea used in EARS.

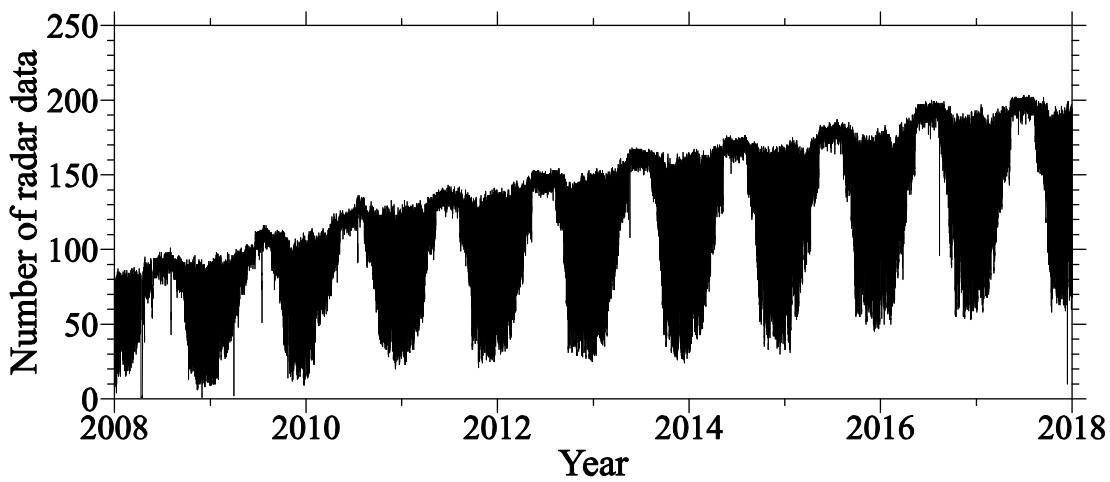

**Fig. 8** Counts of the ground-based radar stations used in EARS.

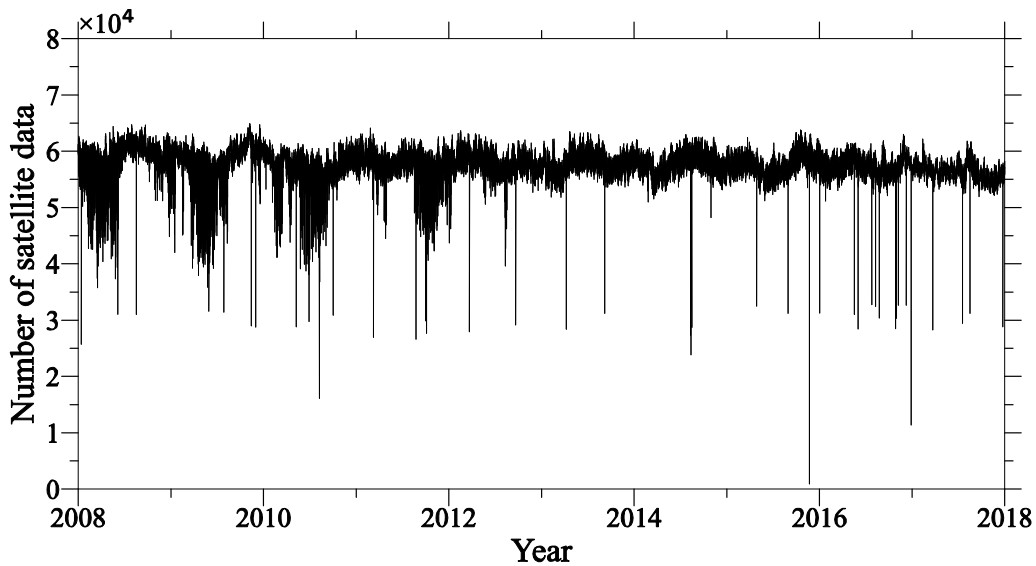

**Fig. 9** Counts of cloud-derived wind vectors from the Fengyun-2 (FY-2) geostationary meteorological satellite observations, which are used in the EARS.

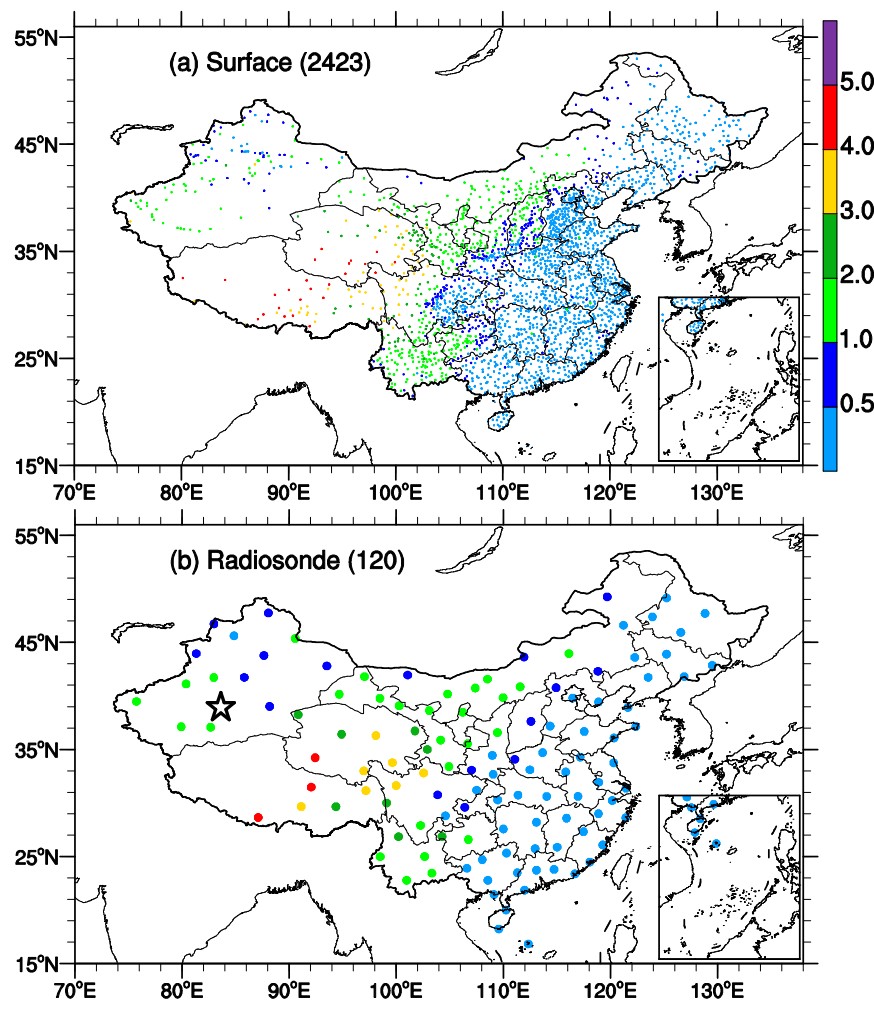

**Fig. 10** Spatial distributions of selected (a) surface and (b) radiosonde stations for verification. The number in parenthesis is the observational count. The color of the dot indicates terrain (km). In (b), the star denotes the location of a radiosonde observational field experiment at the central Taklimakan Desert, China.

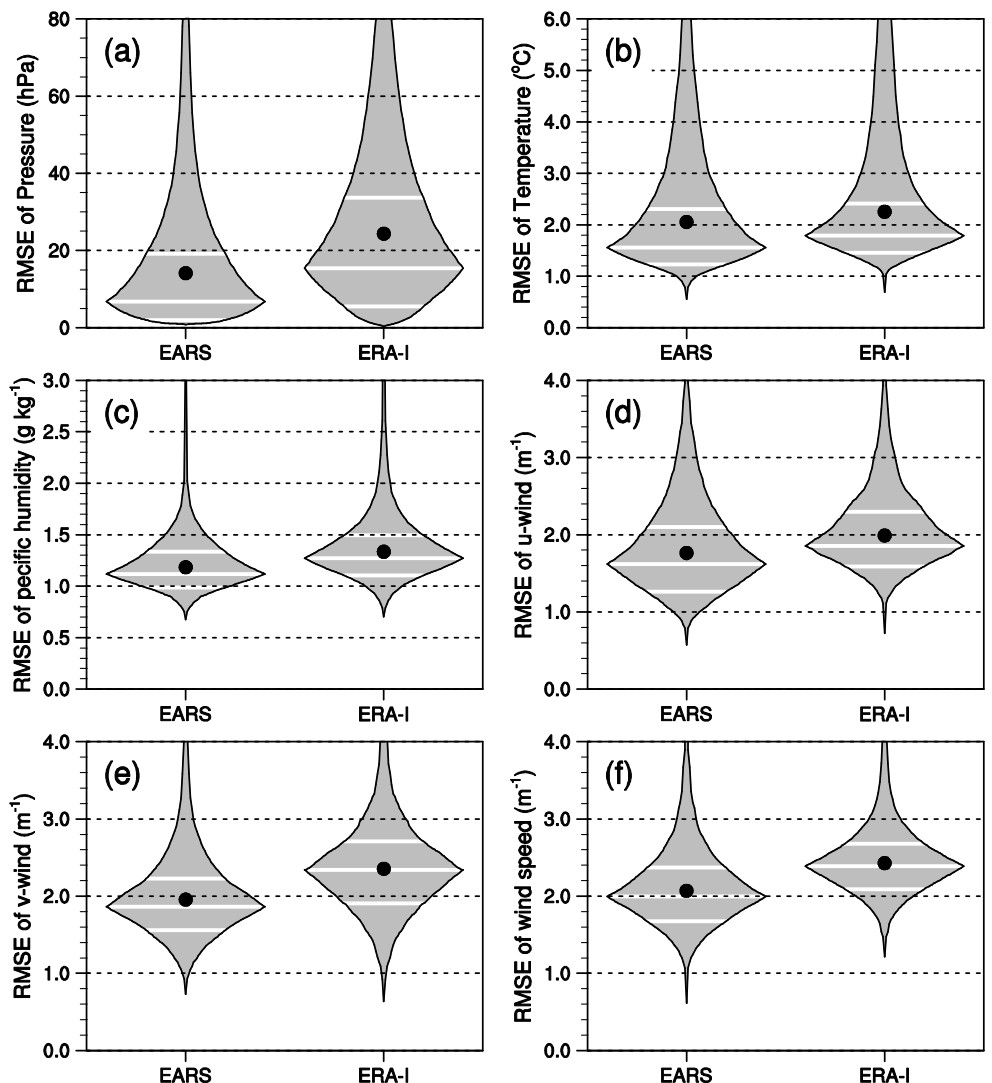

**Fig. 11** Comparison of averaged root mean square error (RMSE) between EARS and ERA-Interim in terms of surface meteorological variables: (a) pressure ($P$; hPa), (b) temperature ($T$; °C), (c) specific humidity ($Q$; g kg$^{-1}$), (d) zonal wind ($U$; m s$^{-1}$), (e) meridional wind ($V$; m s$^{-1}$), and (f) wind speed ($WS$; m s$^{-1}$). The black dot denotes the averaged value of each category; and the horizontal white lines indicate the 25th, 50th

(median), and 75th percentiles. See text for details.

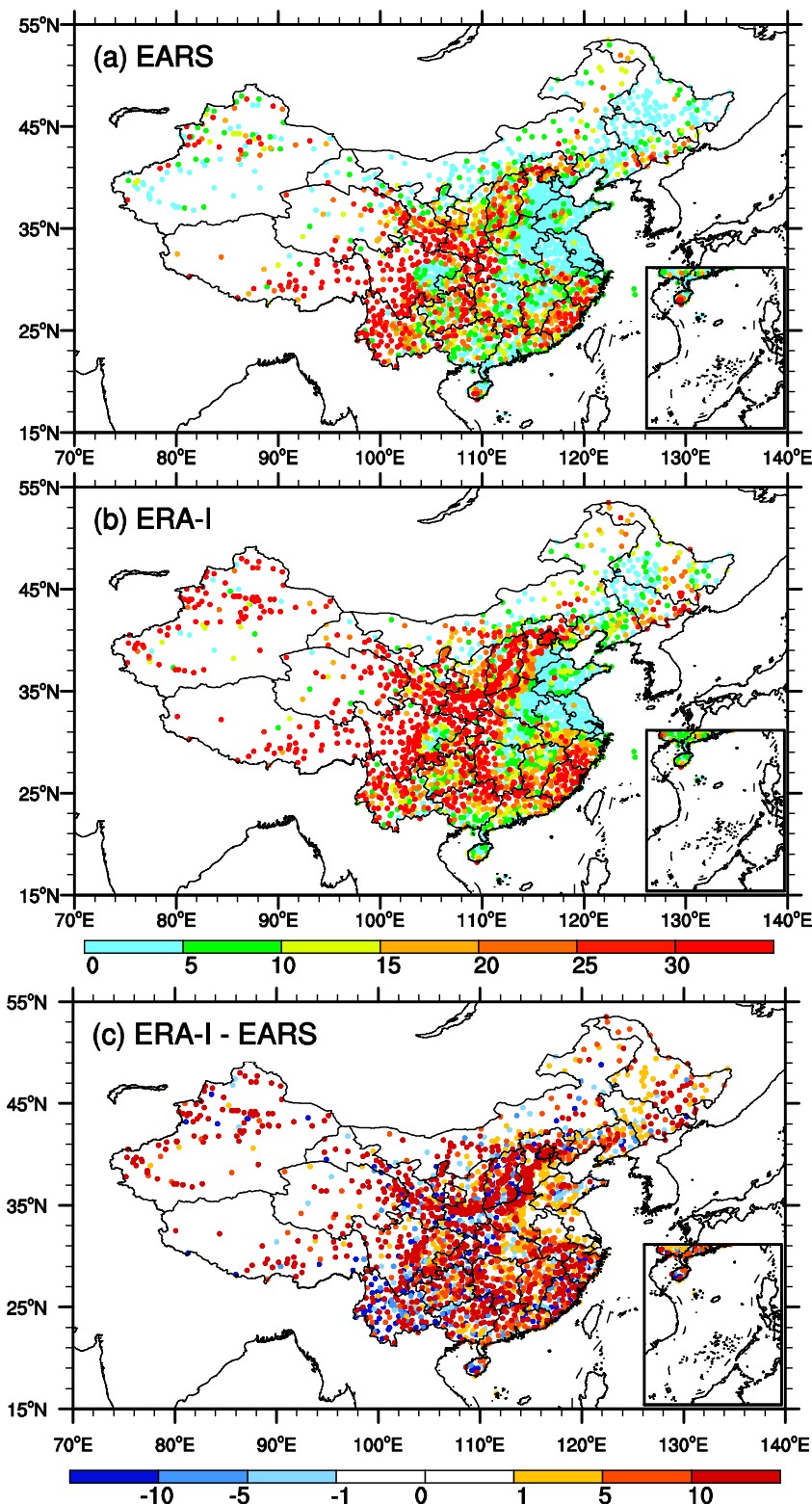

**Fig. 12** Spatial distributions of averaged RMSE of surface pressure (*P*; hPa) from (a) EARS and (b) ERA-Interim; and (c) their differences (ERA-I minus EARS).

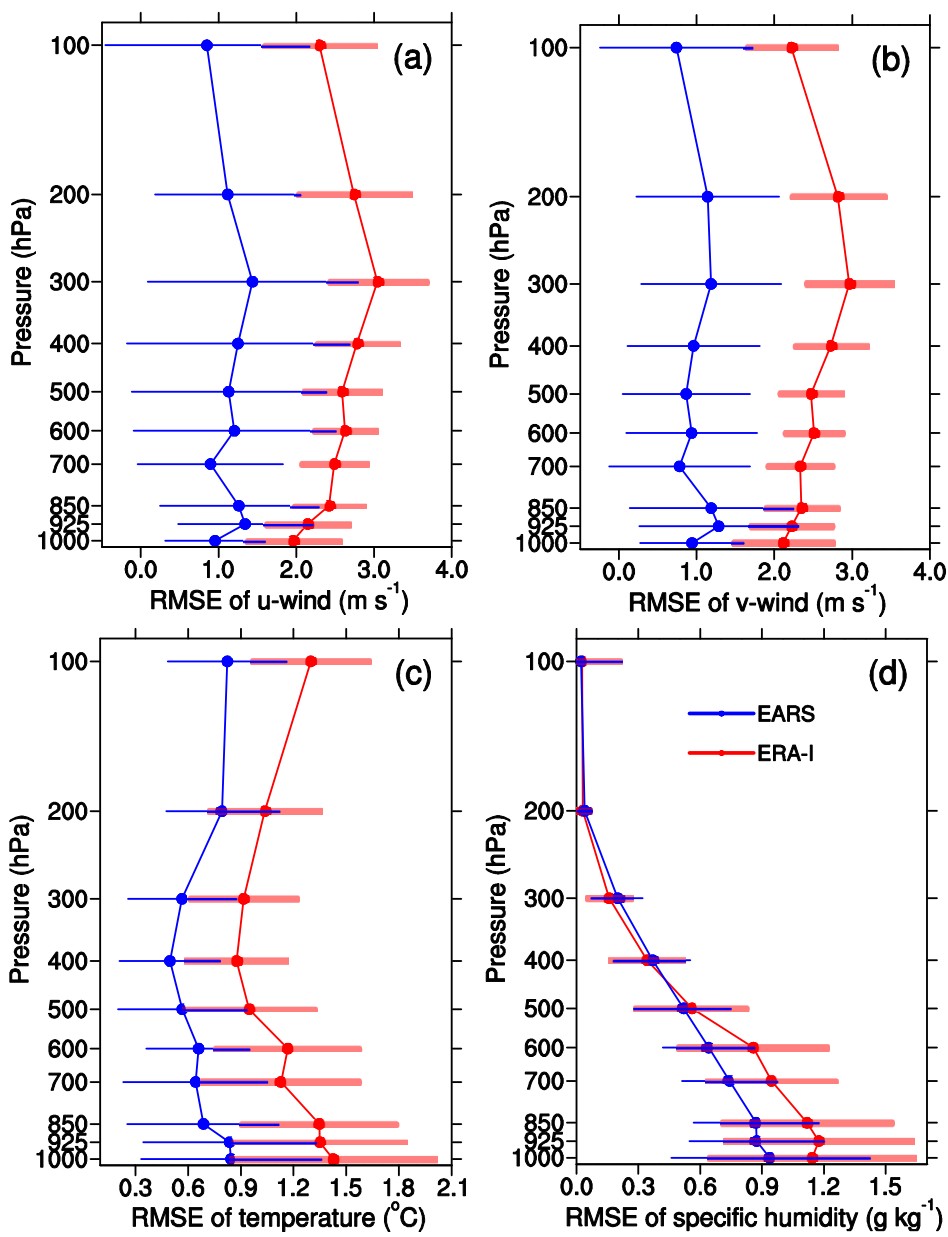

**Fig. 13** Comparison of averaged RMSEs of EARS (blue) and ERA-Interim (red) at different levels for (a) u-wind ($U$; m s$^{-1}$), (b) v-wind ($V$; m s$^{-1}$), (c) temperature ($T$; °C), and (d) specific humidity ($Q$; g kg$^{-1}$). The standard deviation of RMSE at each level is marked by a horizontal line.

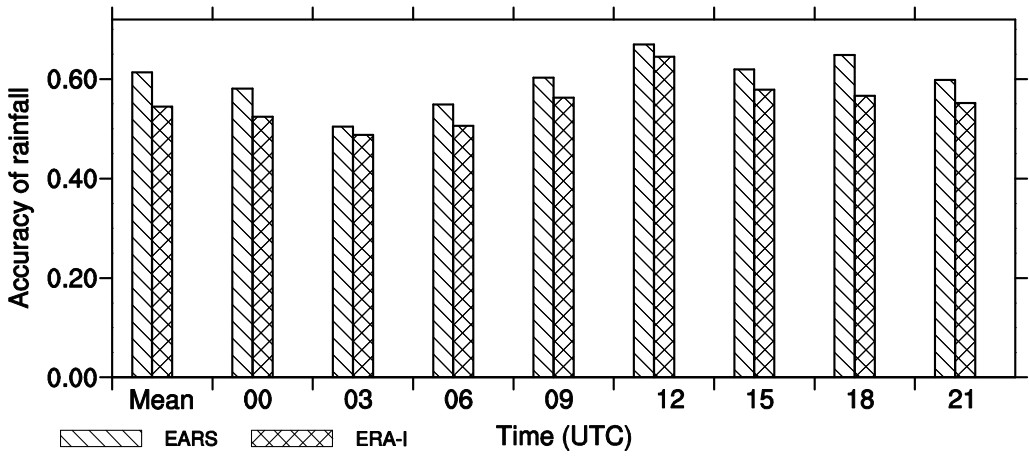

**Fig. 14** Comparison of the accuracy of 3-h accumulated rainfall between EARS
and ERA-Interim in different episodes. Mean denotes the averaged value for all times.

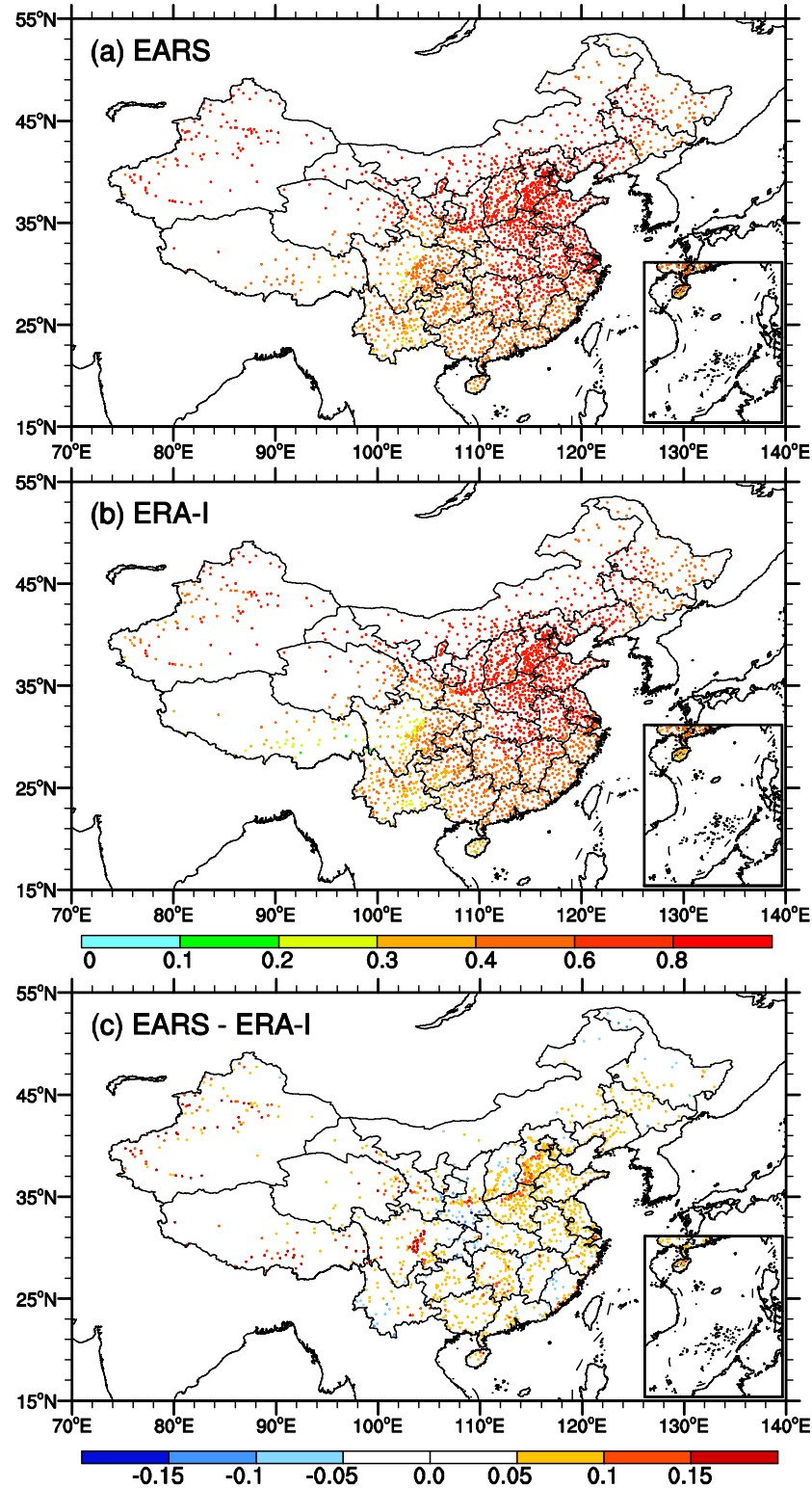

**Fig. 15** Spatial distributions of averaged rainfall accuracy of (a) EARS, (b) ERA-Interim, and (c) their difference (EARS minus ERA-I).

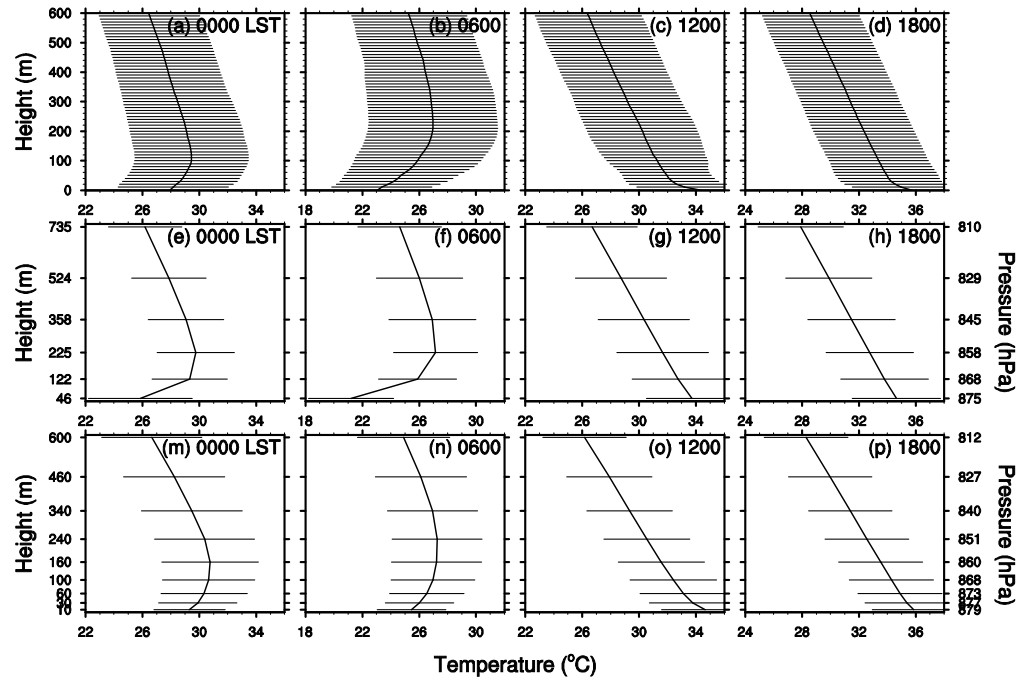

**Fig. 16** Diurnal transitions of the averaged temperature (°C) in lower levels from 0000 to 1800 local standard time (LST, = UTC + 6) in July 2016: (a) Observations (Obs), (b) EARS, and (c) EAR-I. The standard deviation at each level is marked by a horizontal line.

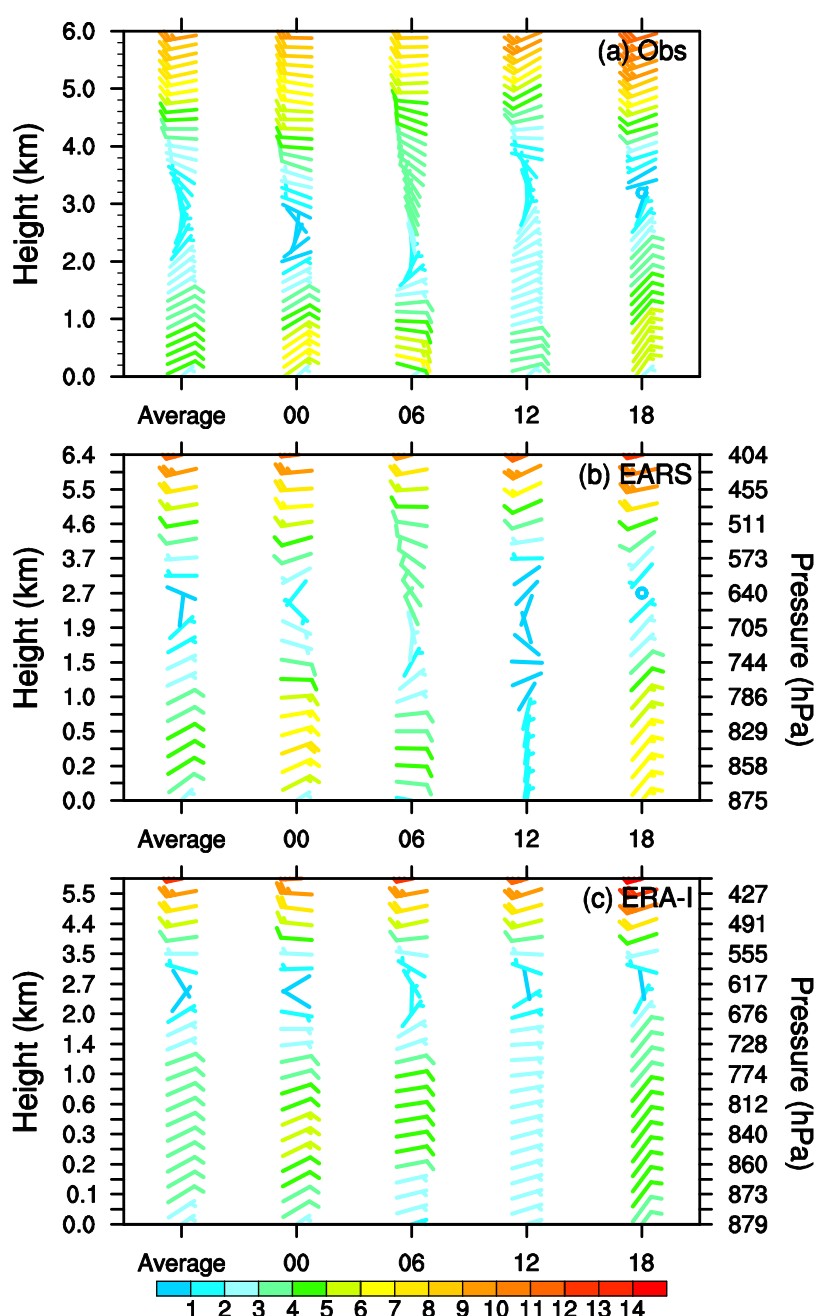

**Fig. 17** Profiles of monthly-averaged horizontal wind barbs in July 2016: (a) observations (Obs), (b) EARS, and (c) EAR-I. A full wind barb denotes 4 m s$^{-1}$; and the shading indicates horizontal wind speed (m s$^{-1}$). Average denotes the total average of all times. Each column corresponds to the average profiles at 0000, 0600, 1200, and 1800 LST.