# Peer review of "East Asia Reanalysis System (EARS)"

_Earth System Science Data, 2022_

## Referee Comment (RC2)

**[General comments]**

Yin et al. developed the East Asia Reanalysis System (EARS) and constructed a 39–year (1980–2018) reanalysis data over East Asia via the system with multi-source observations assimilated. The EARS and used observations are described in detail and principal work is conducted to validate the reliability of reanalysis datasets. This work is necessary and the conducted data has important potential applications for regional weather and climate studies. This manuscript is generally in a good shape. However, several minor revisions are still required before publication, listed as follows.

**[Minor comments]**

(1) The EARS covers a large domain. However, the observations out of China were not used in the validation. Although the results are reasonable and representative, it is advisable to give a detailed explanation in the text.

(2) Did the authors compare EARS with other regional and/or global reanalysis data, such as ERA5, CFSR, JMR, and others? This may be beyond the scope of this paper as the main purpose of this paper is to present EARS and preliminary results. If not, please specify this issue, which may encourage readers to conduct potential associated work.

(3) Given the present results, the EARS datasets are encouraging and promising. This paper is to report the progress of the project. I suggest the authors try to share all the EARS data to the public as soon as possible for potential applications.

(4) Lines 100-102: changing "*intending to produce a high-resolution 100 regional atmospheric reanalysis dataset for East Asia, with high quality for mesoscale weather system study and regional climate analysis*" to "*intending to produce a high-resolution 100 regional atmospheric reanalysis dataset with high quality for mesoscale weather system study and regional climate analysis over East Asia* ".

(5) Line 71: Please provide the horizontal resolution of China's first generation of global atmospheric reanalysis (CRA40) for general information.

(6) Line 176: changing "regular" into "conventional".

(7) Lines 306 and 311: missing "the" before RMSE.

(8) Line 324: modifying "*that WRF downscaling at a high resolution has significant performance gains in downscaling*" to "*significant performances have been gained in WRF downscaling at a high resolution*".

(9) Line 397: Please provide references for "*previous studies and with operational predictions*".

---

## Author Comment (AC1)

**Response to Referee #1's (Dick Dee) comments**

On behalf of all the co-authors, I would like to thank the reviewer, Referee #1, Dick Dee, for his thoughtful and constructive comments which helped us to improve our study. We have responded to the comments in the attached file.

This paper describes the East Asia Reanalysis System (EARS), a regional reanalysis covering all of East Asia, with 3-hourly products provided at a horizontal resolution of 12 km and 74 levels in the vertical. The paper covers the methodology, the use of observations, and a variety of performance aspects. The paper is well organized and well written, with explanations in clear language.

In recent years, CMA has made great strides in developing an ambitious reanalysis program, which has already delivered a global atmospheric reanalysis (CRA40) and now also a unique regional reanalysis product. As the authors point out, EARS is the first regional reanalysis covering all of East Asia. This fills an important gap.

According to the paper, all reanalysis data as well as many of the observations used will be accessible via the China Meteorological Data Service Centre at data.cma.cn. (I was not yet able to find the data when I tried during this review). It is very gratifying and good news for the global reanalysis community that CMA is making their data products and observation data available.

The work on observations that has been done in preparation of the ERAS production is significant and potentially very valuable. As the authors point out, many of the observations have not been used before, either for global numerical weather production or for reanalysis. It is very good news for the scientific research community if CMA is indeed able to share these data openly. It would be good to have more information (possibly in a separate paper) describing the observations and their quality control.

Overall, I think this is a good paper about an important dataset that can be highly valuable for large groups of users around the world. I have many questions and

suggestions to the authors for additional work, but I don't think there is need for a major revision. My recommendation is therefore to publish after minor revision.

**Answer:** Thank you very much for your comments. We are glad to see that our dataset is beneficial to the community. We have thoroughly considered the concerns and will revise the manuscript accordingly.

**Here are my comments and questions about the details:**

If I understand correctly, the background fields used for the reanalysis are WRF short forecasts, which are initialized from ERA-Interim data, and using ERA-Interim data for lateral boundary conditions. There is a 6-h spin-up. Do you have any diagnostics (or have you investigated) the size of the spin-up for different variables, and whether this spin-up depends on the interval (e.g. 6-h vs. 12-h vs. 24-h)? Spin-up can be especially significant for precipitation and cloud, especially because the model used to generate ERA-Interim data is very different from the WRF model.

**Answer:** We agree with you that it is important to determine the spin-up period as it depends on model configuration, such as time-step, horizontal resolution, and boundary conditions. Following the WRF model user guide, a spin-up period was calculated on the basis of d(surface pressure)/dt or d(mu)/dt in the present work (Fig. 1). One can see, d(surface pressure)/dt decreases rapidly in the first 120 min, then tends to be stable after nearly 300 min. Given the result, a 6-h spin-up period was used.

[Figure]

**Fig. 1** Evolution of d(surface pressure)/dt with the WRF model integration time

As you stated, the WRF model bias increased with increasing integration time length (Fig.2) because of the gaps between the ERA-Interim data and the WRF model. Further sensitivity tests indicated that the mismatch of soil moisture and soil temperature resulted in the main bias.

In terms of the WRF model spin-up processes and bias, a 6-h spin-up period was cut and the WRF model integrates 12 h for each run.

[Figure]

**Fig. 2** Evolution of model absolute bias during the period from 01 to 30 June 2016. Mean, Min, and Max denote the mean, maximum, and minimum biases for different forecast length, respectively. The blue dashed lines represent the values of mean, maximum, and minimum biases of ERA-Interim during this period. Std denotes the standard deviation, which suggests the range of bias variation.

Can you provide statistics of the analysis increments (defined as: EARS analysis minus WRF forecast)? This will help to expose biases in the system, due to biases in the model and/or in the observations.

**Answer:** Thank you for providing these ideas. In early experiments before long-term runs, we made comparisons between WRF forecast and analysis. There were slight differences in surface fields because surface observations were nudged during model integration, rather than assimilated at the analysis moment. As for the upper-level variables, RMSEs were significantly reduced, owing to the assimilation of upper-level observations using a 3D-VAR method. After that, we paid no attention to the difference between the WRF forecast and analysis. Since the differences between the WRF forecast and analysis help to expose biases in the system, statistical analyses in this area will be carried out with special attention.

Can you provide more information about the nudging scheme used to introduce surface observations? Does the scheme depend on estimates of uncertainty of the observations?

**Answer:** Thanks for this suggestion. In the revision, we have added the suggested content to the manuscript in Subsection 2.1.

"More specifically, observation nudging is a type of four-dimensional data assimilation (FDDA) wherein artificial tendency terms are introduced during the model integration (Reen, 2016). Since it is applied at every time step, nudging is a continuous form of data assimilation. Therefore, observations in the model integration time window can be ingested. Generally speaking, the differences between the WRF model and observation are utilized to create an innovation. Then, the innovation is multiplied by various factors and added to model tendency equations. It should be noted that observation nudging is affected by the uncertainty of the observations. Therefore, surface observations are strictly quality controlled by the OBSGRID module (Wang et al., 2017)."

References:

Reen, B.: A brief guide to observation nudging in WRF, https://www2.mmm.ucar.edu/wrf/users/docs/ObsNudgingGuide.pdf (last access 17 February 2023), 2016.

Wang, W., Bruyère, C., Duda, M., and Dudhia, J.: User's Guides for the Advanced Research WRF (ARW) Modeling System, http://www2.mmm.ucar.edu/wrf/users/docs/user_guide_V3/contents.html (last access 17 February 2023), 2017.

Can you provide more information about the quality control steps used to prepare the input observations, especially the older observations recovered from analogue sources?

**Answer:** We already provided more information about the quality control steps in Subsection 2.2 as follows:

"Observations were greatly improved by combining datasets from various data sources, especially the observations over China. Firstly, the duplicate (in time and location) data reports were merged. Secondly, all the ground-based observations, were checked by climatic outliers and variation ranges. Besides, internal consistency between meteorological elements and temporal consistency were also carried out. Moreover, soundings were examined based on hydrostatic assumption, temperature lapse rate, and horizontal wind shear."

Can you provide details on any bias corrections applied to the observations?

**Answer:** In the present study, all observations were assumed to be unbiased like most data assimilation systems. Consequently, the observations were assimilated directly without any bias correction employed.

What kind of automated quality control is applied in GSI for the upper-air analysis? Do you have any statistics on the rejection rates etc.?

**Answer:** To the best of our knowledge, simple automated quality control is applied in GSI because of high-quality observations in BUFR/prepBUFR. Specifically, only the gross check for each data type is performed. Because of this, observations were quality controlled using OBSPROC (provided by WRF) before being written in

prepBUFR format. As it is addressed in the WRF user guide (Wang et al., 2017), the purpose of OBSPROC is to:

- ✓ Remove observations outside the specified temporal and spatial domains;
- ✓ Re-order and merge duplicate (in time and location) data reports;
- ✓ Retrieve pressure or height based on observed information using the hydrostatic assumption;
- ✓ Check multi-level observations for vertical consistency and superadiabatic conditions;
- ✓ Assign observation errors based on a pre-specified error file;
- ✓ Write out the observation file to be used by WRFDA in ASCII or BUFR format.

After strict quality control, almost all the observations passed GIS gross check.

References:

Wang, W., Bruyère, C., Duda, M., and Dudhia, J.: User's Guides for the Advanced Research WRF (ARW) Modeling System, http://www2.mmm.ucar.edu/wrf/users/docs/user_guide_V3/contents.html (last access 17 February 2023), 2017.

Can you provide more information about the characteristics the background error covariances used in the GSI analysis?

**Answer:** It is well known that background error covariance (BE) plays an important role in three-dimensional variational assimilation. In this work, the NMC method was used to build BE based on one-month continuous WRF simulations in each season (Fig. 3). Compared with the default BE (Fig. 3d), more details can be obtained from the built BE. The finer the latitude band is used for statistics, the more detailed the features are (Fig. 3a-c). However, from idealized and real cases simulations, the newly built BE provided poorer performance than the default BE. Therefore, we used the NAM background error covariance with scale factor adjustment in this study.

[Figure]

**Fig. 3** Properties of background error covariance (BE). (a)-(c) are calculated from 5 °, 1 °, and 0.1 °latitude bands, respectively. (d) is the default option in the GSI.

Can you provide more information about the assimilation of radar data? Has there been any pre-processing of the radar data?

**Answer:** We already provided more information about the assimilation of radar data in Subsection 2.2 as follows:

In the present work, radar reflectivity was ingested by the way of cloud analysis, while the radial wind was not assimilated at present as more work is required. The cloud analysis module in the GSI came from the Advanced Regional Prediction System (ARPS) (Hu and Xue, 2007), and can be further traced to the Local Analysis and Prediction System (LAPS) (Albers et al., 1996). The radar basic data was provided by CMA. All the radar basic data were quality controlled, such as removing isolated non-meteorological echoes and ground clutter. After quality control, all radar observations at the same time are utilized to generate mosaic products in BUFR format, which can be inserted into the GSI cloud reanalysis module.

References:

Hu, M., and M. Xue, 2007: Implementation and evaluation of cloud analysis with WSR-88D reflectivity data for GSI and WRF-ARW. *Geophysical Research Letters*, **34**, doi:10.1029/2006GL028847.

Albers, S. C., J. A. McGinley, D. L. Birkenheuer, and J. R. Smart, 1996: The Local Analysis and Prediction System (LAPS): Analyses of Clouds, Precipitation, and Temperature. *Weather and Forecasting*, **11,** 273-287, doi:10.1175/1520-0434(1996)011<0273:TLAAPS>2.0.CO;2.

Many of the validation results in the paper refer to the improvements in EARS relative to ERA-Interim. Those are mostly good results, but they are not very surprising given the higher resolution and use of many additional observations. I think that it would be very useful to show more diagnostics that focus on the use of observations specifically, such as time series of observation-minus-background statistics. These can be very informative and can be used to identify issues and problems with the observations and/or the data assimilation scheme, that could possible be addressed in a future reanalysis.

**Answer:** Thank you for this great suggestion. GSI outputs detailed diagnostic files which provide useful information to diagnose potential benefits and problems. To date, We have not yet carried out detailed statistical work and only sampled some basic characteristics, such as cost and gradient function (Fig. 4), and scatter plot of observation-minus-background (O-B). Keeping this suggestion in our mind, detailed statistical analyses will be carried out in the future.

[Figure]

**Fig. 4** Evolution of cost function (left) and the norm of gradient of cost function (right) in the first outer loop (top) and the second outer loop (bottom) plotted vs assimilation iteration number at 1200 UTC 01 July 2014.

Fig 5: Is this for a single sounding? What does the shift signify? There is no description of the x-axis.

**Answer:** Thanks for your kind reminder. Yes. We just took the sounding at Beijing station (54511) at 0000 UTC 1 July 2016 as an example. All the soundings in China were processed with the same approach. The two profiles are perfectly overlapped except for newly added observation points. To avoid overlaying the two data points, we have shifted the IGRAv2 profile slightly to the left. We further illustrated this in the figure caption.

Fig 11: I don't understand the grey shapes in this figure.

**Answer:** The grey fill is to highlight the box-percentile plots, although the black border is readable.

---

## Author Comment (AC2)

**Response to Reviewer 2's comments**

We would like to thank Reviewer, Referee #2, for his/her thorough review and constructive comments that have helped improve the quality of our manuscript. To address Reviewer 2's comments, we have made revisions in the text. Our point-by-point responses to Reviewer 2's comments are given below. For your convenience, we have also uploaded a version with tracked changes.

Yin et al. developed the East Asia Reanalysis System (EARS) and constructed a 39–year (1980–2018) reanalysis data over East Asia via the system with multi-source observations assimilated. The EARS and used observations are described in detail and principal work is conducted to validate the reliability of reanalysis datasets. This work is necessary and the conducted data has important potential applications for regional weather and climate studies. This manuscript is generally in a good shape. However, several minor revisions are still required before publication, listed as follows.

**Answer:** We appreciate you very much for your kind words about our manuscript. Your positive and constructive comments and suggestions are valuable in improving the quality of the manuscript.

**[Minor comments]**

(1) The EARS covers a large domain. However, the observations out of China were not used in the validation. Although the results are reasonable and representative, it is advisable to give a detailed explanation in the text.

**Answer:** We already provided an explanation in Subsection 2.3 as follows:

"It should be noted that the present validation is based on the observations from China Meteorological Administration (CMA). Although the EARS covers a large area, only limited observations out of China were obtained by the Global Communication System (GTS). Comparatively speaking, the density of observations is much higher in China than that out of China. Besides, the performance of observations in China is at a comparable level because of the same (at least equivalent) observational instruments and methods. Moreover, the observations in China were quality controlled using the same methods. Therefore, the observations in China were used in the validation. We welcome more validation from others with observations outside of China as much as possible."

(2) Did the authors compare EARS with other regional and/or global reanalysis data, such as ERA5, CFSR, JMR, and others? This may be beyond the scope of this paper as the main purpose of this paper is to present EARS and preliminary results. If not, please specify this issue, which may encourage readers to conduct potential associated work.

**Answer:** We agree that this would be an interesting question, and we have further discussed it in the last paragraph. We are fully occupied with EARS development and data generation. The EARS data was verified against both surface and sounding observations. The results were also compared with its parent一the ERA-Interim. At

present, comparisons with other global reanalyses are on our schedule but progress is slow (e.g., Yang et al., 2022). As far as we know, the assessment of reanalysis data is a complex and systematic task. Therefore, we expect more scholars to evaluate EARS data from different aspects, such as the performance in reproducing weather systems (e.g., Gong et al., 2022), daily variation in precipitation (e.g., Li et al., 2017), and others, as well as comparisons among different reanalysis (e.g., Yang et al., 2022).

Gong, Y., S. Yang, J. Yin, S. Wang, X. Pan, D. Li, and X. Yi, 2022: Validation of the Reproducibility of Warm-Season Northeast China Cold Vortices for ERA5 and MERRA-2 Reanalysis. *Journal of Applied Meteorology and Climatology*, **61,** 1349-1366, doi:10.1175/JAMC-D-22-0052.1.

Li, J., T. Chen, and N. Li, 2017: Diurnal Variation of Summer Precipitation across the Central Tian Shan Mountains. *Journal of Applied Meteorology and Climatology*, **56,** 1537-1550, doi:10.1175/JAMC-D-16-0265.1.

Yang, L., X. Liang, J. Yin, Y. Xie, and H. Fan, 2023: Evaluation of the Precipitation of the East Asia regional reanalysis system mainly over mainland China. *International Journal of Climatology*, doi:10.1002/joc.7940.

(3) Given the present results, the EARS datasets are encouraging and promising. This paper is to report the progress of the project. I suggest the authors try to share all the EARS data to the public as soon as possible for potential applications.

**Answer:** Thank you very much for your kind words about this work. We agree with you that the large volume of data prevents them from being shared effectively. The database format is GRIB version 1 and the total volume of the data files is 54.6 TB. As the total data exceeds the volume that could be provided by freely Zenodo (Yin et al., 2022), we will put the data on the CMA Data-as-a-Service platform (http://data.cma.cn/). However, storing data on the CMA platform requires a certain amount of time for application and approval. We will complete the sharing of the EARS data as soon as authorization is granted. Relevant information will also be updated in Zenodo (Yin et al., 2022).

Yin, J., X. Liang, Y. Xie, F. Li, K. Hu, L. Cao, F. Chen, H. Zou, F. Zhu, X. Sun, J. Xu, G. Wang, Y. Zhao, and J. Liu, 2022: East Asia Reanalysis System (EARS) [Data set]. *Zenodo*, doi:https://doi.org/10.5281/zenodo.7404918.

(4) Lines 100-102: changing "*intending to produce a high-resolution regional atmospheric reanalysis dataset for East Asia, with high quality for mesoscale weather system study and regional climate analysis*" to "*intending to produce a high-resolution regional atmospheric reanalysis dataset with high quality for mesoscale weather system study and regional climate analysis over East Asia* "

**Answer:** Revised accordingly.

(5) Line 71: Please provide the horizontal resolution of China's first generation of global atmospheric reanalysis (CRA40) for general information.

**Answer:** Revised accordingly, i.e., "although China's first generation of global atmospheric reanalysis (CRA40) was released recently, with a horizontal resolution of approximately 34 km and a temporal resolution of 6 h".

(6) Line 176: changing "regular" into "conventional".
**Answer:** Done.

(7) lines 306 and 311: missing "the" before RMSE.
**Answer:** Done.

(8) Line 324: modifying "*that WRF downscaling at a high resolution has significant performance gains in downscaling*" to "*significant performances have been gained in WRF downscaling at a high resolution*".
**Answer:** Revised accordingly.

(9) Line 397: Please provide references for "*previous studies and with operational predictions*".
**Answer:** As suggested by the reviewer, we have added more references to support this idea (e.g., Mao et al., 2010; Zhang et al., 2021; Zhao et al., 2018).

Zhang, J., T. Zhao, Z. Li, C. Li, Z. Li, K. Ying, C. Shi, L. Jiang, and W. Zhang, 2021: Evaluation of Surface Relative Humidity in China from the CRA-40 and Current Reanalyses. *Adv. Atmos. Sci.*, **38,** 1958-1976, doi:10.1007/s00376-021-0333-6.

Zhao, S., W. He, and Y. Jiang, 2018: Evaluation of NCEP-2 and CFSR reanalysis seasonal temperature data in China using detrended fluctuation analysis. *International Journal of Climatology*, **38,** 252-263, doi:10.1002/joc.5173.

Mao, J., X. Shi, L. Ma, D. P. Kaiser, Q. Li, and P. E. Thornton, 2010: Assessment of Reanalysis Daily Extreme Temperatures with China's Homogenized Historical Dataset during 1979–2001 Using Probability Density Functions. *Journal of Climate*, **23,** 6605-6623, doi:10.1175/2010JCLI3581.1.

---

## Author Response (AR2)

**Author's Response**

Dear Dr. David Carlson,

On behalf of all co-authors, I appreciate you and the reviewers for reviewing our paper (entitled "East Asia Reanalysis System (EARS)", essd-2022-429) and providing valuable and insightful comments that led to possible improvements in the present version. We have carefully considered the comments and tried our best to address every one of them, and the manuscript has been revised accordingly. For your convenience, we have also uploaded a version with tracked changes. An item-by-item reply to the Reviewers is shown as follows.

I look forward to hearing from you soon.

Sincerely yours,

Dr. Jinfang Yin

April 14, 2023

**Response**

1. Line 165: Users might need citation for the 2014 assessments?

**Answer:** Thank you very much for your kind suggestion. Before generating long-term reanalysis, one-year (2014) test reanalysis data was generated and validated. Although the test data supported our subsequent work, we did not expend the effort to release it. Readers may refer to the reports of Yin et al. (2019) and Liang et al. (2020) for some results.

2. Line 174 (and later, Line 189, line 193, 194, 232, etc.): as both reviewers noted, with previously-unavailable data mentioned as strong feature of ERAS, they really need more information about QC for newly-assimilated observations? In addition to reference in individual sections (surface, radiosonde, radar), include a summary statement about unique aspects and challenges of additional local data? Show readers/users how you have added benefit while incorporating standard QC functions? Perhaps in conclusion section? Review ESSD guidelines (at https://www.earth-syst-sci-data.net/10/2275/2018/) to ensure you have met all expectations.

**Answer:** Thank you. It's a good question. We already explained in Section 4 (Lines 476-484) as follows.

"An important feature of the EARS is the use of a large number of observations from CMA. Compared with IGRA version 2, more than 33 operational radiosonde observations over China were used. Besides, more radiosonde vertical-level observations were included by merging logs of old records. Moreover, radiosonde observations from field experiments over China were also employed. A large number of aircraft observations and surface (over land and sea) hourly observations over China were utilized. Note that only a small portion of the observations has been shared in the GTS. Another characteristic is the application of over 200 Doppler radar observations and the cloud-derived wind vector datasets from Fengyun-2 geostationary meteorological satellites. "

3. Line 178 (): readers will need assurance of availability at CMA!

**Answer:** At present, the website is maintained by the China Meteorological Administration (CMA). Readers are free to register and download most of the data. Although all data is placed online, some data needs (e.g. radar observations) require further requests before they can be accessed. It should be emphasized that the use of the data is under the regulations of the CMA.

4. Line 242: Need clarification / changed language here?

**Answer:** Thank you for your kind reminders. We revised the sentence as follows:
"Note the number of radar observations shows obvious seasonal various because some radars were switched off in cold seasons due to the absence of weather processes."

5. Lines 290 and following (again at line 458): comparison here to data from single radiosonde station during July 2016. How differs from earlier comparison mentioned for 2014?

**Answer:** Generally speaking, the results in the present study are close to those in earlier comparison of the test reanalysis data in 2014. This is understandable as the East Asian reanalysis system (EARS) has been frozen since then. The radiosonde observations were carried out in July 2016 in the central Taklimakan Desert, Xinjiang Uygur Autonomous Regions, China. The central Taklimakan Desert is far from other observation sites and the data were not applied to any reanalysis systems. Therefore, we took the radiosonde data as independent observations.

6. Line 388: Need clarification / changed language here?

**Answer:** Thank you very much for your kind suggestion. We revised the sentence as follows:
"The RMSEs for $Q$ decrease rapidly with increasing height and approach zero near 200 hPa."

7. In a few figures (e.g. Fig 7), panels (a, b) mentioned in legend do not appear in graphics? Please assure that readers find accurate representations?

**Answer:** Thank you for the kind reminder. Fig. 7 has been updated. We have double-checked the figures to ensure that all figures are correctly labeled.

8. Users will need clearer guidance about what data products to find and use from Zenodo vs CMA. Provide clarification in data availability section?

**Answer:** We have revised data availability section accordingly to make it clearer. In order to effectively share the EARS reanalysis data, several ways were taken to promote the data. A Digital Object Unique Identifier (DOI: https://doi.org/10.5281/zenodo.7404918) was applied based on the *Zenodo*. One of the advantages is that we can provide details of EARS in English, and can keep them up to date. However, the total volume of the EARS data exceeds the volume that could be provided by Zenodo. Therefore, we have to host the data at the CMA Data-as-a-Service platform (http://data.cma.cn/). It should be noted that only a general description of the ERAS can be given on the CMA website in Chinese. Therefore, users can obtain comprehensive and up-to-date information about EARS and sample data in *Zenodo*, and all data can be downloaded from the CMA Data-as-a-Service platform (http://data.cma.cn/).